# Gating and noelin clustering of native Ca²⁺-permeable AMPA receptors

Chengli Fang[1], Cathy J. Spangler[1], Jumi Park[1], Natalie Sheldon[1,2], Laurence O. Trussell[1,3] & Eric Gouaux[1,2 ✉]

AMPA-type ionotropic glutamate receptors (AMPARs) are integral to fast excitatory synaptic transmission and have vital roles in synaptic plasticity, motor coordination, learning and memory[1]. Whereas extensive structural studies have been conducted on recombinant AMPARs and native calcium-impermeable (CI)-AMPARs alongside their auxiliary proteins[2–5], the molecular architecture of native calcium-permeable (CP)-AMPARs has remained undefined. Here, to determine the subunit composition, physiological architecture and gating mechanisms of CP-AMPARs, we visualize these receptors, immunoaffinity purified from rat cerebella, and resolve their structures using cryo-electron microscopy (cryo-EM). Our results indicate that the predominant assembly consists of GluA1 and GluA4 subunits, with the GluA4 subunit occupying the B and D positions, and auxiliary subunits, including transmembrane AMPAR regulatory proteins (TARPs) located at the B′ and D′ positions, and cornichon homologues (CNIHs) or TARPs located at the A′ and C′ positions. Furthermore, we resolved the structure of the noelin (NOE1)–GluA1–GluA4 complex, in which NOE1 specifically binds to the GluA4 subunit at the B and D positions. Notably, NOE1 stabilizes the amino-terminal domain layer without affecting gating properties of the receptor. NOE1 contributes to AMPAR function by forming dimeric AMPAR assemblies that are likely to engage in extracellular networks, clustering receptors in synaptic environments and modulating receptor responsiveness to synaptic inputs.

AMPARs have fundamental roles in fast excitatory neurotransmission and synaptic plasticity[1,6,7]. They are classified into two functional subtypes on the basis of their calcium permeability; receptors that incorporate the RNA-edited GluA2 subunit are largely CI-AMPARs[8–10], and are predominant in the mammalian brain[11], whereas CP-AMPARs, which typically lack the edited GluA2 subunit, exhibit unique electrophysiological properties—such as calcium permeability, inward rectification and polyamine sensitivity—all of which influence synaptic function and plasticity[1,9,12]. CP-AMPARs are pivotal for rapid synaptic responses and are dynamically regulated during forms of synaptic plasticity such as long-term potentiation and long-term depression[7,13], and have been implicated in various neurological disorders[12].

AMPARs are tetrameric assemblies composed of subunits GluA1 to GluA4 and exhibit a modular architecture comprising three layers: the extracellular N-terminal domain (ATD), the extracellular ligand-binding domain (LBD) and the transmembrane domain (TMD)[2]. The ATD is involved in receptor assembly, subunit interactions and interactions with synaptic proteins, such as neuronal pentraxin and noelin[14,15], which in turn are implicated in synapse formation and receptor anchoring[16]. The LBD is the site of neurotransmitter binding, initiation of channel gating and desensitization. Structural studies have shown that the LBDs can adopt a spectrum of 'desensitized' conformations, from simple rupture of the LBD domain 1 (D1)–D1 interface in a twofold-symmetric conformation to large-scale rearrangements of the LBDs to an approximately fourfold-symmetric arrangement during prolonged glutamate exposure[17]. The TMD forms the ion channel pore, determining ion selectivity and conductance, and its function is further modulated by TARPs and CNIHs, which contribute to receptor trafficking, gating and pharmacology[18–24]. Together, the ATD, LBD and TMD form a Y-shaped architecture in the resting state of GluA2-containing CI-AMPARs, stabilized by dimer-of-dimer interactions within the ATD layer of the receptor assembly[2].

By contrast, recombinant CP-AMPAR GluA1 homomers have a conformationally heterogeneous ATD layer that alters the swapping of domains from the ATD to the LBD layer and affects channel gating kinetics, suggesting that in CP-AMPARs, the ATD layer has an important role in receptor gating[25]. To understand the overall architecture, subunit arrangement, auxiliary protein composition and function of native CP-AMPARs, we isolated them directly from the cerebellum, a brain region that contains an abundance of CP-AMPARs.

## Cerebellar CP-AMPARs are mainly GluA1/A4

In the cerebellum, CP-AMPARs are constitutively expressed in basket cells[22], stellate cells[26] and oligodendrocyte precursor cells[27,28], where they contribute to synaptic integration and neuronal excitability. CP-AMPARs are also distributed throughout Bergmann glia, and are activated by ectopic glutamate release onto those cells[29,30].

[1]Vollum Institute, Oregon Health and Science University, Portland, OR, USA. [2]Howard Hughes Medical Institute, Oregon Health and Science University, Portland, OR, USA. [3]Oregon Hearing Research Center, Oregon Health and Science University, Portland, OR, USA. ✉e-mail: gouauxe@ohsu.edu

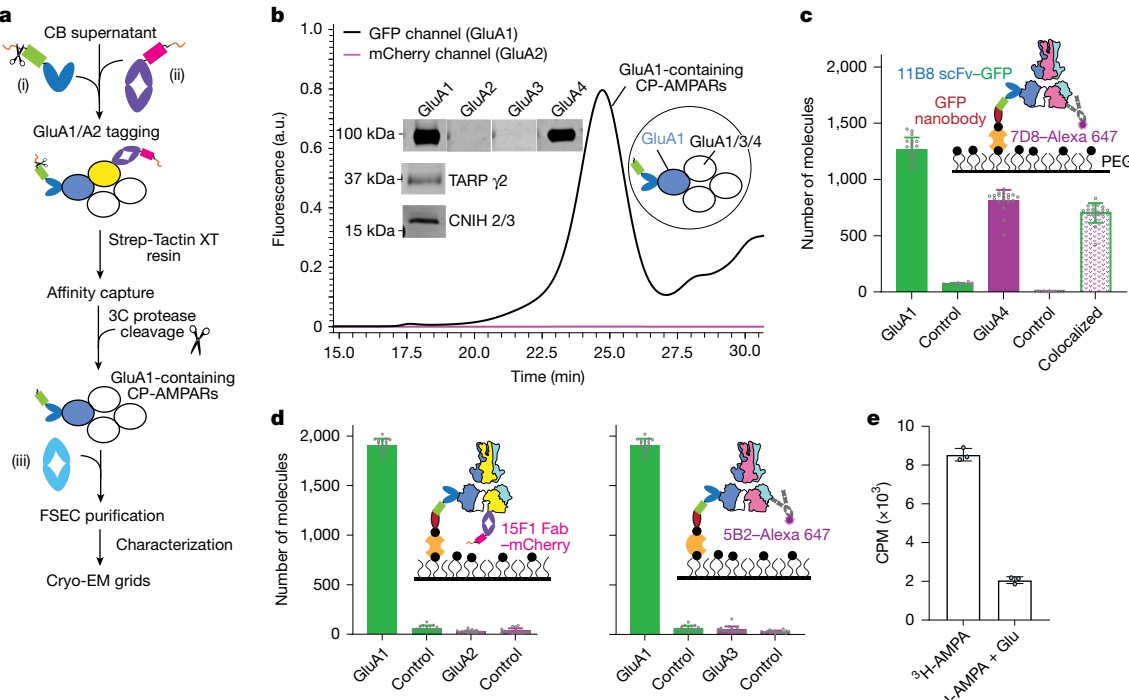

**Fig. 1 | Isolation of non-GluA2 CP-AMPARs from the cerebellum. a**, Flow chart showing isolation of CP-AMPARs from rat cerebellum (CB). The use of 3C–Twin-Strep tagged anti-GluA1 11B8 scFv–GFP (i) and Twin-Strep tagged anti-GluA2 15F1 Fab–mCherry (ii) enabled GluA1 and GluA2 to be captured simultaneously. The 3C protease selectively releases 11B8–GluA1-containing AMPARs from the resin. Pale blue oval (iii), anti-GluA3 5B2 Fab; dark blue oval, GluA1; yellow oval, GluA2; white oval, GluA1, GluA3 or GluA4. **b**, Representative FSEC profile of the native CP-AMPARs. The black trace shows 11B8–GluA1 signals in the GFP channel. The pink trace, which represents the mCherry channel, shows no detectable 15F1–mCherry GluA2 signal. The protein purification and FSEC were independently reproduced three times with consistent results. Insets,

western blot for CP-AMPARs and a cartoon of CP-AMPARs. The western blot was independently repeated twice with consistent results. The uncropped blot can be found in Supplementary Fig. 3. a.u., absorbance units. **c**,**d**, SiMPull analysis of the isolated CP-AMPARs, examining GluA1 colocalization with GluA4 (**c**) or with GluA2 and GluA3 (**d**). Inset, schematic of the SiMPull assay. PEG, polyethylene glycol. Data are from $n = 17$ images and are shown as mean ± s.d. See Methods for a description of the control experiments. **e**, Single-point radioligand-binding assay of $^3$H-AMPA binding to the CP-AMPARs. Nonspecific binding was estimated by adding 10 mM 'cold' glutamate (right column). Data are mean ± s.d. ($n = 3$, where $n$ denotes the number of parallel measurements). CPM, counts per minute.

To investigate whether GluA1-containing AMPARs from the cerebellum are devoid of GluA2, we performed single-molecule pull-down (SiMPull) experiments[31] utilizing supernatant from rat cerebellar tissue applied to chambers coated with the GluA1 monoclonal antibody 4H9. Subsequently, fluorophore-labelled antibodies for GluA1 (11B8–Alexa488), GluA2 (15F1–Alexa555) and GluA4 (7D8–Janelia 646) were applied to the chambers[5] (Extended Data Fig. 1a,b and Supplementary Figs. 1 and 2). We observed that the abundance of GluA4 was significantly higher than that of GluA2, with around 70% of GluA1 colocalizing with GluA4 and only approximately 27% of GluA1 colocalizing with GluA2 and GluA4 (GluA1/A2/A4; Extended Data Fig. 1a–c). This suggests that GluA1-containing AMPARs in the cerebellum predominantly exist as GluA1–GluA4 (GluA1/A4) heteromers (around 40%) that are devoid of the GluA2 subunit.

To purify CP-AMPARs along with their auxiliary proteins, we implemented a rapid purification strategy (Fig. 1a). We engineered a GluA1-specific 11B8 single-chain variable fragment (scFv) fused with GFP and a Twin-Strep tag. A strategically inserted human rhinovirus (HRV) 3C site between the GFP and the affinity tag facilitates the release of the protein from the affinity resin. Furthermore, we developed a GluA2-specific 15F1 Fab fragment, which incorporates mCherry and a Twin-Strep tag but does not contain a protease site. This experimental design enables us to capture GluA1- and GluA2-containing AMPARs, and the 3C protease selectively releases only the GluA1-containing AMPARs from the beads (Fig. 1a). Following fluorescence detection size-exclusion chromatography (FSEC) purification, the GFP channel (GluA1) displayed a distinct peak, whereas no signals were observed

for the mCherry channel (GluA2), consistent with AMPARs devoid of the GluA2 subunit (Fig. 1b). This conclusion was further supported by western blot, mass spectrometry and SiMPull assay results (Fig. 1, Extended Data Fig. 1d–g and Supplementary Fig. 3). We speculate that the isolated GluA1-containing CP-AMPARs are derived primarily from Bergmann glia and Purkinje cells, the former of which express both GluA1 and GluA4 subunits[32]. Using $^3$H-AMPA radioligand-binding assays, we demonstrate that the purified CP-AMPARs possess active agonist binding sites (Fig. 1e).

## Non-stochastic assembly of CP-AMPARs

To elucidate the architecture and subunit arrangement of CP-AMPARs, we characterized the structure of cerebellar GluA1-containing CP-AMPARs in the presence of MPQX, a competitive antagonist that stabilizes receptors in a resting, non-desensitized state (Extended Data Fig. 2 and Extended Data Table 1). In contrast to GluA2-containing receptors that contain a stable ATD layer (Supplementary Fig. 4), 2D classification indicated that some receptor particles contain a conformationally mobile ATD layer, whereas the TMD and LBD layers were more well-resolved (Extended Data Fig. 2c). Multiple rounds of heterogeneous refinement ultimately identified six distinct classes. Classes 1 and 2, which comprise around 50% of the particles and are characterized by well-defined ATD layers, exhibit a Y-shaped conformation, with two 11B8 scFv features at the A and C positions. Additionally, particles in class 1 with the subunits at the B and D positions were untagged by a scFv or Fab domain and thus are GluA4, in accordance with the biochemistry

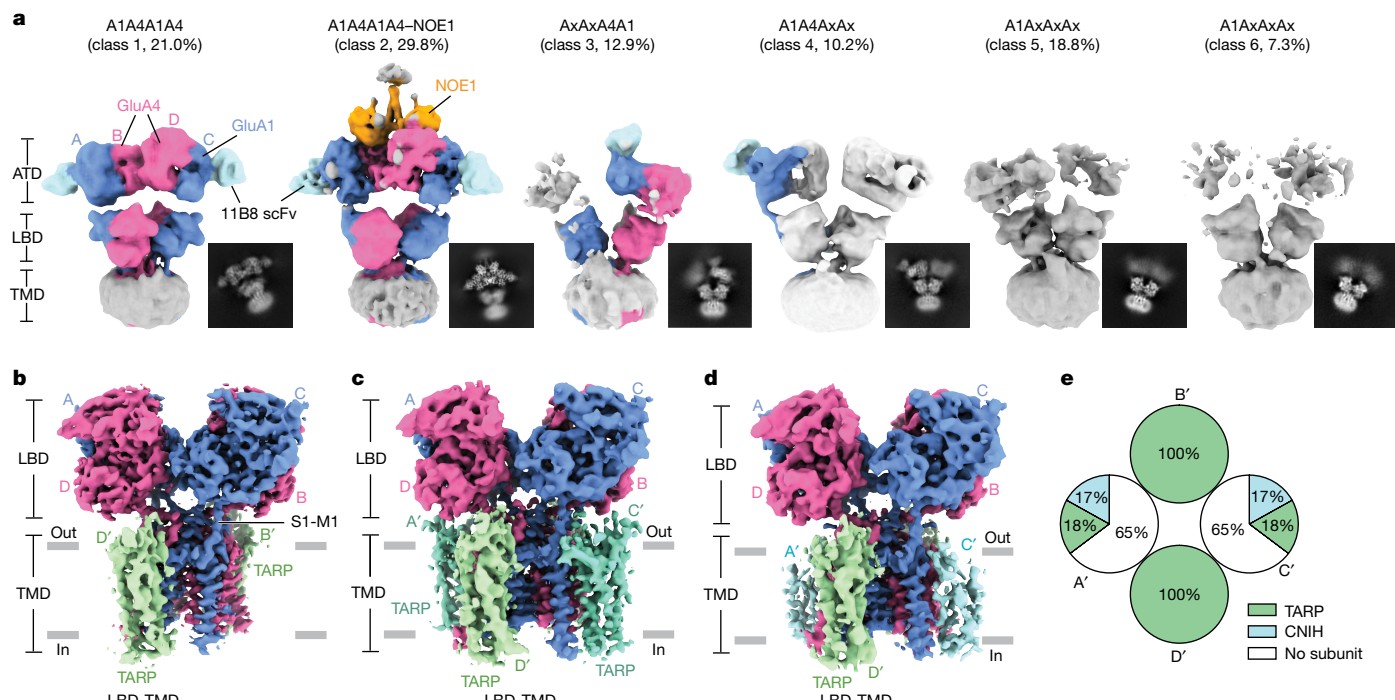

**Fig. 2 | Cryo-EM analysis shows CP-AMPAR subunit arrangement, ATD binding partners and conformation. a**, Representative 3D classes of CP-AMPARs in the closed state. In classes 5 and 6, which have a completely flexible ATD layer, A1 in A1AxAxAx does not represent a subunit at the A or C position but rather indicates pull-down by the GluA1 antibody. **b**, LBD–TMD$_{mix}$ with two TARPs occupying B′ and D′ positions. **c**, LBD–TMD$_{mix}$ with four TARPs occupying A′, B′, C′ and D′ positions. **d**, LBD–TMD$_{mix}$ with two TARPs occupying B′ and D′ positions and 2 CNIHs occupying A′ and C′ positions. **e**, Pie chart showing auxiliary protein distribution as a function of position around the receptor TMD and how the B′ and D′ positions are fully occupied by TARPs.

results (Fig. 1). We categorize class 1 as A1A4A1A4 receptors, using a previously defined nomenclature[33]. By contrast, classes 3 and 4 present only a single, clear ATD dimer, each displaying a unique 11B8 scFv signal at the B or D, or A or C positions. The remaining two classes, classes 5 and 6, demonstrate disordered ATD features (Fig. 2a) and adopt a diverse array of conformations, complicating structural determination and the unambiguous identification of receptor subunits.

In addition to diversity encoded by assembly with multiple receptor subunits, AMPAR function is modulated by auxiliary proteins that influence receptor trafficking, gating, kinetics, pharmacology and ion selectivity[10,18,33,34]. In the cerebellum, the TARP γ2 and γ7 subunits are highly expressed[24,32,35], whereas γ4, γ5, γ8 and CNIH2 are also present[24,35]. To explore the composition and arrangement of auxiliary proteins, we focus on the LBD–TMD layer of particles from all classes, except for those in class 2, because in this class, we observe additional density on the 'top' of the receptor, suggesting that this class represents a distinct group of receptors. Nevertheless, given the high sequence identity among GluA1 to GluA4 subunits, we enhanced the LBD–TMD density map by incorporating all particles from the remaining classes. Focused classification and non-uniform refinement of the LBD–TMD layer identified three distinct classes based on auxiliary subunit arrangements (Fig. 2b–e and Extended Data Figs. 3 and 4). The first class (nominal resolution 3.7 Å) revealed only two auxiliary subunits exhibiting prominent extracellular protrusions at the B′ and D′ positions, with no discernible density features at the A′ and C′ positions. Notably, the absence of auxiliary proteins at the A′ and C′ positions is correlated with weak density for the receptor M2 helices, consistent with previous studies showing that fully occupied auxiliary subunits stabilize M2 helices[36]. We designate this class as LBD–TMD$_{mix-2TARPs}$ on the basis of cryo-electron microscopy (cryo-EM) densities and TARP features (Fig. 2b and Extended Data Figs. 3 and 4f). We infer that the TARP subunits are largely contributed by γ2 on the basis of the western blot and mass spectrometry results. The second class (4.0 Å) contained

four auxiliary subunits at the A′, B′, C′ and D′ positions, all displaying significant extracellular protrusions, and we termed this class LBD–TMD$_{mix-4TARPs}$ (Fig. 2c and Extended Data Figs. 3 and 4d). Despite the overall resolution being 4.0 Å, the transmembrane helices were all well-resolved, including the M2 helices, with clear side-chain signals, particularly glutamine at the Q/R site, which determines $Ca^{2+}$ permeation (Extended Data Fig. 4a–c). The third class (4.0 Å) also contained four auxiliary subunits; however, the densities at the A′ and C′ positions are weak and lacked extracellular protrusions, leading us to hypothesize that these are CNIHs, and we termed this class LBD–TMD$_{mix-2TARPs-2CNIHs}$ (Fig. 2d and Extended Data Figs. 3 and 4e). The transmembrane helices were well-resolved, including the M2 helices and the glutamine at the Q/R site, with clear side-chain signals, similar to LBD–TMD$_{mix-4TARPs}$.

Focused classification of the A1A4A1A4 complexes (class 1), in the context of the LBD–TMD layer, yielded two classes and the arrangement of the auxiliary subunits similar to the above (Extended Data Fig. 5). One class shows two auxiliary proteins at B′ and D′ positions with a distinctive extracellular protrusion, with no discernible density features at the A′ or C′ positions. Another class displayed four auxiliary subunits at the A′, B′, C′ and D′ positions, but weaker density for four transmembrane helices at the A′ and C′ positions without extracellular domain features, thus suggesting that within this class there is a mixture of TARPs and CNIHs at A′ and C′ positions. However, owing to a limited number of particles, we cannot further classify this group (Extended Data Fig. 5).

Overall, we observed that 65% of particles contain 2 TARPs at B′ and D′ positions, 17% contain 2 TARPs at the B′ and D′ positions and 2 CNIHs at the A′ and C′ positions, and 18% contain 4 TARPs at the A′, B′, C′ and D′ positions (Fig. 2e). Across all three classes, around 77% of particles belonged to classes 1 and 5. Because these receptor complexes were isolated from total cerebellar tissue, we speculate that the complexes with four TARPs are localized to the cell membrane and probably comprise the receptor assemblies associated with most electrophysiological recordings[37], although further experiments are required to

conclusively determine the auxiliary subunit constellation of plasma membrane-localized receptor assemblies. Previous structural studies of GluA2-containing CI-AMPARs have shown defined patterns of auxiliary subunit occupancy (Supplementary Fig. 4) that include four TARP γ2 subunits in GluA2 homomers[36,38], two TARP γ8 subunits in GluA1/A2 heteromers[39], and various combinations of TARP γ8 and CNIH2 occupying distinct positions in native and recombinant receptors[40,41]. In these complexes, TARPs are consistently found in the B′ and D′ positions, whereas there is greater auxiliary subunit heterogeneity at the A′ and C′ positions, similar to our native CP-AMPAR structures. We infer that TARPs preferentially occupy B′ and D′ positions because these sites provide ample space for its extracellular domain, which promotes stronger interactions with receptors than CNIHs.

## The ATD layer adopts multiple conformations

Recombinant homomeric GluA1 receptors have a conformationally heterogeneous ATD layer, even when the receptor is stabilized in a resting, non-desensitized state, and the dynamical behaviour of the ATD layer is linked to receptor gating[25]. We thus aimed to determine the conformational ensembles of native CP-AMPARs and subsequently probe relationships between ATD conformation and receptor function. We found that native CP-AMPARs comprising A1A4A1A4 subunit assemblies have a distinct conformation of the ATD layer compared with other classes (Fig. 2a). To further investigate the A1A4A1A4 complex, we carried out an additional round of heterogeneous refinement and identified two distinct structural classes: one with an ordered ATD layer and another with a blurry ATD layer (Extended Data Figs. 3 and 5a–c). Despite the heterogeneity within the ATD layer of the latter class, we found that it corresponds to an A1A4A1A4 complex, as confirmed by 2D class averages and 3D reconstruction analyses (Extended Data Fig. 5a–c). These findings suggest that GluA1/A4 assemblies can adopt either an ordered or disordered ATD layer when GluA4 occupies the B and D positions. Indeed, we substantiated this observation by showing that recombinant homomeric GluA4 assemblies exhibit both ATD conformations (Supplementary Fig. 5). Thus, we speculate that the dimer–dimer interface with the ATD layer of GluA4 receptors is different from that of GluA1 and GluA2 receptors. In GluA1 receptors, the ATD layer is entirely disordered, whereas in the GluA2 receptors, it is stable and 'intact', and in the GluA4 receptors, it exhibits 'intermediate' stability.

Inspection of the dimer–dimer interface in the ATD layer provides molecular explanations for the differences in stability between AMPARs composed of different subunits. In GluA2 receptors, the interface is stabilized by a network of residues surrounding helices αF, αG and αH[2]. In GluA2, the cation–π interaction between F231 (αH) and R172 (αF) is essential for stabilizing the interface. Similarly, in GluA4, H234 and R175, which correspond to F231 and R172 in GluA2, can form a cation–π interaction under neutral conditions (Extended Data Fig. 5b). However, H234 is surrounded by a local acidic environment created by residues D171, E179, Q204, S207 and E230. This environment may lead to the protonation of H234, potentially disrupting its interaction with R175. As a result, GluA4 exhibits both ordered and disordered ATD layers, perhaps driven by the protonation state of H234 (Extended Data Fig. 5b). Previous studies suggest that the GluA2 ATD dimer–dimer interface modulates proton-mediated desensitization recovery, primarily through H208 (αG)[42]. By contrast, GluA4 contains two histidine residues (H211 and H234, corresponding to GluA2 H208 and F231, respectively), we thus speculate that GluA4 may exhibit greater sensitivity to pH fluctuations than GluA2.

## NOE1 forms a complex with CP-AMPARs

Inspection of the density features in class 2 showed that this group of particles has a stable, Y-shaped ATD layer and a density feature protruding from the 'top' of the ATD that is neither a scFv nor a Fab fragment (Fig. 2a). To define the molecular identity of this density feature, we used mass spectrometry analysis, and found abundant counts for NOE1 (also known as olfactomedin 1 (OLFM1)) and noelin-3 (NOE3), an olfactomedin domain-containing secreted glycoprotein (Fig. 3a and Extended Data Fig. 1d). Of note, the olfactomedin domains of NOE1 or NOE3 align exceptionally well with this density, enabling us to conclude that class 2 represents a noelin–AMPAR complex. The mass spectrometry results indicate that NOE1 is more abundant than NOE3 and analysis of the cryo-EM densities associated with glycosylation patterns and residue side chains is consistent with NOE1 being the predominant species (Extended Data Fig. 4g). NOE1 is a highly expressed secreted protein in the brain and has been shown to co-purify with AMPAR complexes from mouse brain tissue[24,43]. Notably, NOE1-knockout mice exhibit a low survival rate[43]. NOE1 consists of a tetramerization domain, an elongated coiled-coil domain, a short coiled-coil region and an olfactomedin domain, which dimerizes via the short coiled-coil domain and interacts with the ATD of the receptor (Fig. 3). Mass spectrometry identified peptides covering all domains (residues 37–457) of NOE1, suggesting that we had purified the full-length NOE1 in complex with CP-AMPARs. Nonetheless, we note that NOE1 might stabilize the CP-AMPAR complex, thereby making it more abundant in the purified preparation than in the complex in vivo.

To investigate the structure and composition of the NOE1–AMPAR complex, we performed focused refinement on the ATD layer, resulting in a 3.2 Å resolution map that enabled us to conclusively identify the subunit bound to NOE1 (Fig. 3b) as GluA4 (Extended Data Fig. 4h). Consequently, the receptors in class 2 are A1A4A1A4 receptors, in which A4 occupies the B and D positions and A1 is located at the A and C positions. Notably, the NOE1 dimer binds in a twofold-symmetric manner at the 'top' of one AMPAR, with the olfactomedin domain interacting primarily with ATDs at the B and D positions, and only a few contacts to the ATDs at the A and C positions (Fig. 3b,c and Extended Data Fig. 4i). Each NOE1 protomer interacts with one of the ATD dimers, involving an interface area of approximately 779 Å$^2$. The majority of this interface occurs between the GluA4 subunit and NOE1, with each NOE1 protomer forming an interface area of around 595 Å$^2$ with one GluA4 ATD and approximately 184 Å$^2$ with one GluA1 ATD (Fig. 3b,c). The focused classification and refinement of the LBD–TMD layer yielded similar results to the non-NOE1–A1A4A1A4 complex, resulting in a 4.2 Å resolution map that revealed four auxiliary subunits at the A′ to D′ positions (Fig. 3b).

Structural investigations have demonstrated that NOE1 forms a disulfide-linked dimer of dimers with a characteristic V-shaped architecture[44] (Fig. 3a). However, only a small region (the olfactomedin domain) interacts with AMPARs. To definitively validate the interaction between NOE1 and AMPARs, we designed a NOE1 construct that removes the tetramerization domain and the long coiled-coil domain, but includes five residues from GCN4, a classic coiled-coil protein, to stabilize dimerization of the short coiled-coil domain (Fig. 3a and Supplementary Fig. 6). FSEC analysis shows that this construct forms a dimer in the absence of a reducing agent (Supplementary Fig. 6b); we refer to this dimeric NOE1 construct as dNOE1. Further FSEC and bio-layer interferometry analyses demonstrated that dNOE1 binds homomeric GluA4 with high affinity (dissociation constant ($K_d$) ≈ 9.4 nM) (Extended Data Fig. 6a,b). Alanine substitutions of GluA4 residues (Y21A, R25A, H47A, D49A, N50A, S65A and R69A) whose side chains interact with NOE1 altered the elution time, further confirming their role in the GluA4–NOE1 interaction (Fig. 3d and Extended Data Fig. 6j). Additionally, a pull-down assay using dNOE1 and rat brain tissue revealed a significant peak at the position of the AMPAR peak in FSEC (Extended Data Fig. 6f,g). Mass spectrometry analysis confirmed that this peak corresponds to the AMPAR complex, and incubation with 11B8 scFv shifted the peak, indicating that dNOE1 effectively extracts AMPARs from cerebellar tissue (Extended Data Fig. 6h,i).

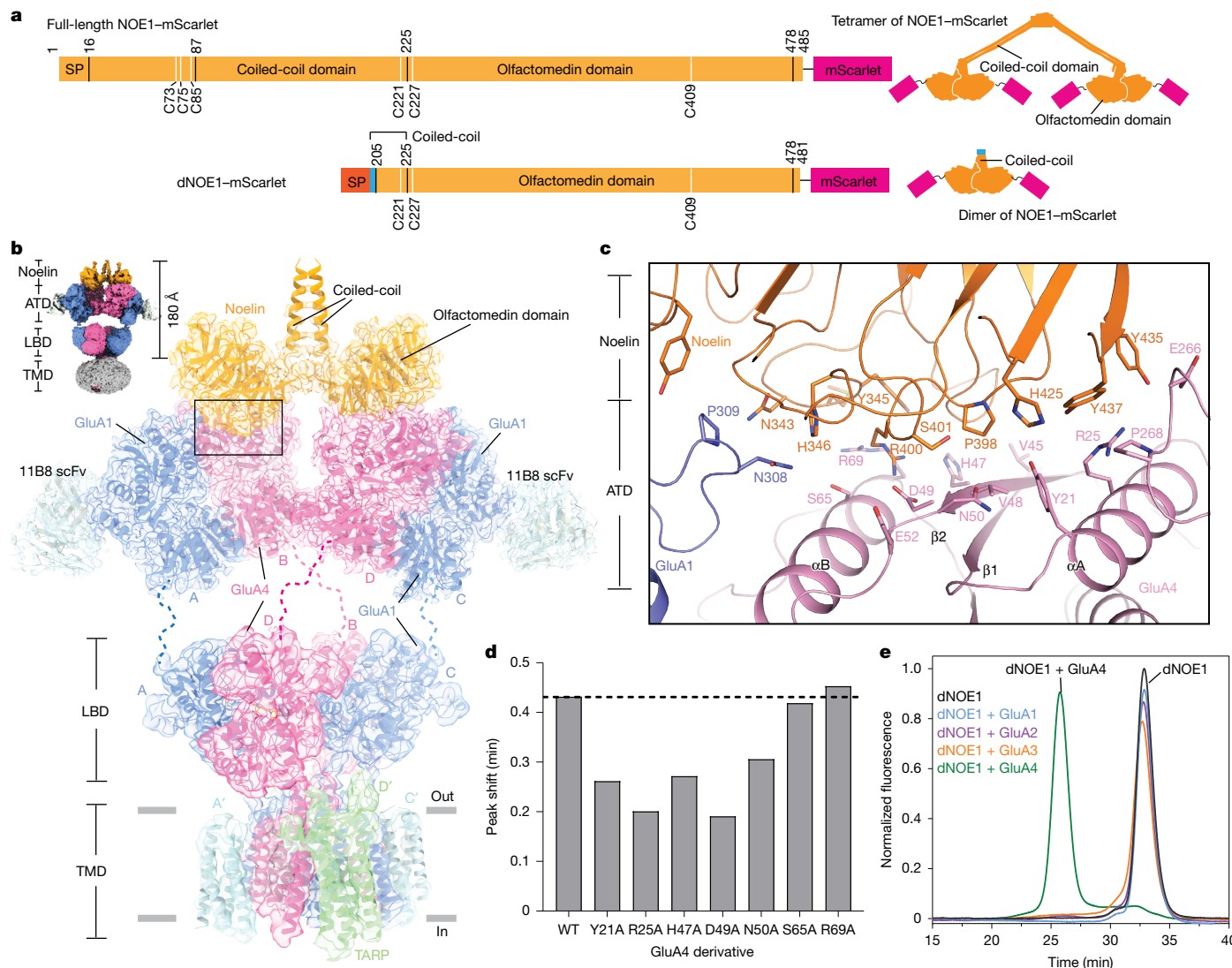

**Fig. 3 | NOE1 primarily binds to the GluA4 subunit of CP-AMPARs.**
**a**, Schematics of full-length NOE1 and dNOE1 domains. The blue bar represents five residues from the coiled-coil of GCN4, which is used to stabilize the dNOE1 dimer. SP, signal peptide. **b**, Main image, a composite cryo-EM map of the ATD and LBD–TMD of the NOE1–AMPAR complex was generated through focused refinement on the ATD layer and the LBD–TMD layer. The individual ATD and LBD–TMD models were fitted in the map. Top left, overall map of NOE1–AMPAR complexes. **c**, The interface between NOE1 and the receptor, highlighting key residues. The structure was visualized using PyMOL 3.1. **d**, GluA4 mutations affect binding to dNOE1–mScarlet, as measured by FSEC peak shift experiments (see Methods and Extended Data Fig. 6j). **e**, FSEC profiles of 5 nM dNOE1–mScarlet (black trace) or recombinant GluA1 (200 nM, blue), GluA2 (200 nM, purple), GluA3 (200 nM, orange) or GluA4 (150 nM, green) with dNOE1–mScarlet, showing mScarlet fluorescence in buffer containing MPQX and (*R*, *R*)-2b. Only GluA4 receptors are shifted.

## NOE1 binds to GluA4-containing AMPARs

Previous studies have shown that the multitude of receptor assemblies containing all AMPAR subunits—including those that lack the ATD, such as GluA1-ΔATD and GluA2-ΔATD, but not GluA4-ΔATD—robustly co-purify with NOE1 (ref. 43). We also observed that NOE1 binds specifically to the ATD of the GluA4 subunit, with alanine substitutions further confirming this interaction. To address this discrepancy, we separately incubated recombinant homomeric GluA1, GluA2, GluA3 or GluA4 receptors with dNOE1. Of these, only the homomeric GluA4 receptor shifted dNOE1 to a higher molecular weight (Fig. 3e), suggesting preferential binding of NOE1 to the GluA4 subunit. This finding was further corroborated by SiMPull assay[31] for full-length NOE1 with GluA1 to GluA4 (Extended Data Fig. 6e). Sequence alignment of the NOE1 binding surface across GluA1 to GluA4 revealed that the interacting residues in GluA4 are not conserved in GluA1 and GluA3, whereas GluA2 shares high sequence conservation with GluA4 (Extended Data Fig. 7a

and Supplementary Fig. 7). Structural comparisons of the ATDs from the GluA1/2 (Protein Data Bank (PDB) ID: 7LDD), GluA1/4 (PDB ID: 9NR8) and the noelin–GluA1/4 receptors (PDB ID: 9NR7) revealed the GluA1/2-ATD heterodimer adopts a larger dimer–dimer angle (104.5°) along the dimer-of-dimers interface than in GluA1/A4-ATD (95.7°) and noelin–GluA1/4-ATD (97.9°) (Extended Data Fig. 7b,c). Furthermore, cryo-EM analysis of AMPARs with GluA2 occupying the B and D positions consistently showed an ordered ATD layer and a stable dimer interface, with comparable dimer–dimers angles (Extended Data Fig. 7c). These observations suggest that when NOE1 interacts with AMPAR subtypes containing GluA2, only one protomer can bind to an ATD dimer, resulting in a potentially unstable interaction. However, GluA1/A4 can adopt both ordered and disordered ATD layers when GluA4 occupies the B and D positions. This indicates that the GluA1/4 receptor possesses a conformationally heterogeneous ATD layer, which enables NOE1 to engage with one of the ATD dimers, subsequently stabilizing it through structural rearrangements captured by NOE1 (Extended Data Fig. 7d).

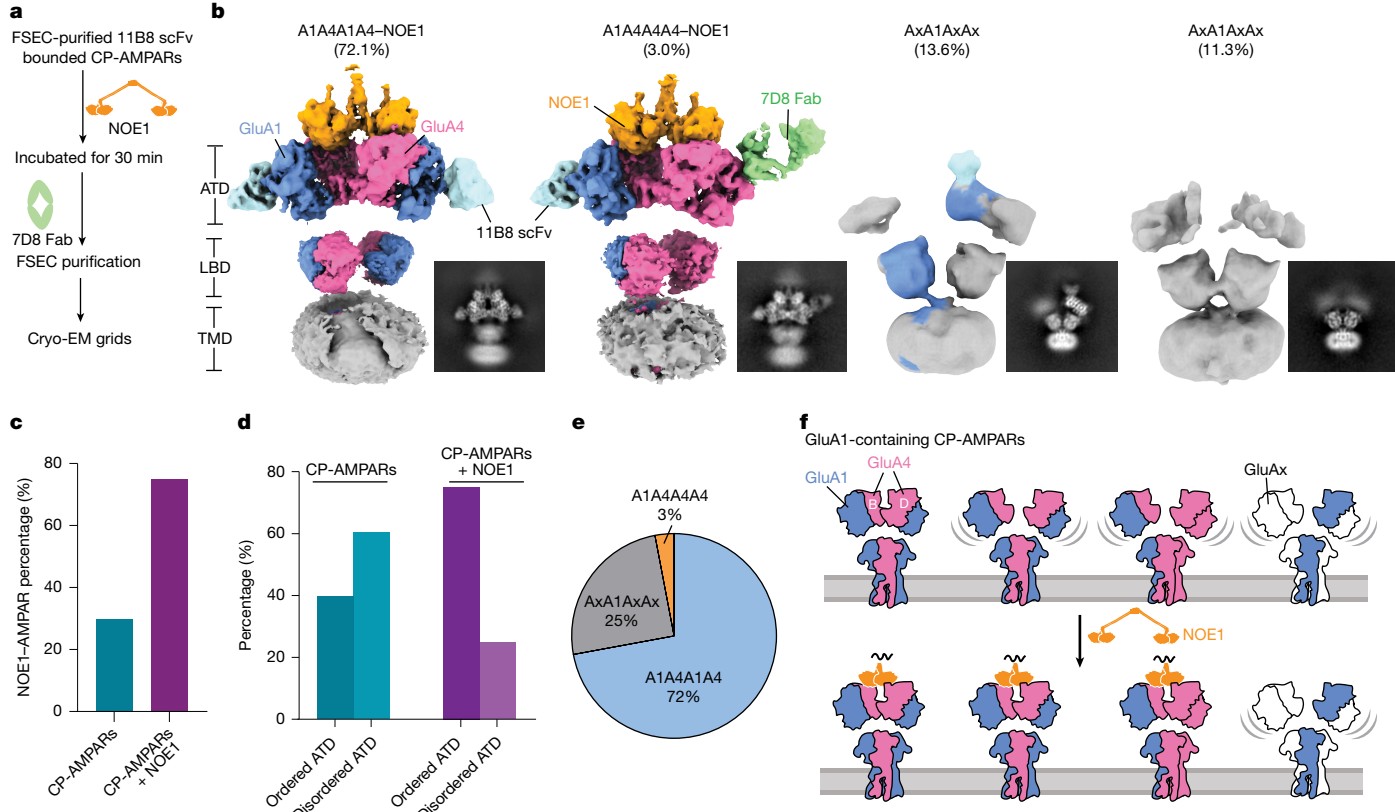

**Fig. 4 | NOE1 restricts the conformational mobility of the ATD layer. a**, Flow chart showing CP-AMPAR–NOE1 complex preparation. **b**, Representative 3D reconstruction of CP-AMPARs with recombinant NOE1. **c**, Bar graph showing how addition of recombinant NOE1 to native CP-AMPARs produces a larger fraction of CP-AMPAR–NOE1 complex. **d**, The percentage of the particles with ordered and disordered ATD layer in the CP-AMPAR dataset with or without recombinant NOE1, showing how the addition of NOE1 confers conformational 'order' to the ATD layer. **e**, The distribution of GluA1-containing CP-AMPAR complexes. **f**, Cartoon showing how NOE1 stabilizes the ATD layer when the GluA4 subunit occupies the B and D positions.

The GluA4 subunit is typically expressed in specific neuronal populations or during specific developmental stages[45], and selective binding to NOE1 may fine-tune synaptic transmission in neural circuits that rely on GluA4 receptors. NOE1 may also regulate synaptic plasticity through interactions with the GluA4 subunit, influencing processes such as long-term potentiation[15,46]. We therefore propose that NOE1 interacts selectively with GluA1/4 receptors in vivo to form complexes that may be important for the function of AMPARs in Bergmann glia, which exhibit high levels of GluA1 and GluA4 (ref. 32). Indeed, AMPARs in Bergmann glia contribute to fine-tuning neuroglial association and neuronal processing[47] by sensing glutamate released from ectopic synaptic sites[30]. We anticipate that the NOE1–GluA1/4 interaction has an important role in stabilizing receptors and that NOE1 potentially interacts with extracellular or presynaptic proteins such as neuritin and brorin[43], ensuring accurate localization at the synapse, thereby enhancing the responsiveness of the receptor to synaptic inputs.

To understand whether the mechanism by which NOE1 binds to CP-AMPARs might extend to CI-AMPARs, we revisited the large ensemble of AMPARs characterized by Zhao and colleagues and noted that one assembly has GluA2 and GluA4 subunits at the B and D positions, respectively[5]. Because the NOE1 binding surface is conserved between GluA2 and GluA4 subunits, we speculate that NOE1 may bind to AxA2AxA4 subtypes. To investigate this hypothesis, we purified GluA2-containing receptors from mouse brain tissue using an antibody to L21-32R in the carboxy-terminal domain of GluA2. Notably, NOE1 induced a shift in GluA2-containing AMPARs, albeit less pronounced than the one induced in CP-AMPARs, resulting in an extended peak (Extended Data Fig. 6c,d). We suggest that NOE1 can bind to GluA2-containing AMPARs, such as AxA2AxA4 receptors, thereby endowing NOE1 with the ability to

influence the structural and functional organization of both CP-AMPARs and CI-AMPARs.

## NOE1 limits ATD layer mobility

The ATD is involved in the trafficking and anchoring of AMPARs to the synapse[16,48] and we speculate that the conformational integrity of the ATD layer may be important for these activities. Given that approximately 30% of the particles were bound to NOE1 in our original dataset and displayed an ordered ATD, we suggest that NOE1 stabilizes the ATD layer when GluA4 occupies the B and D positions. In our preparation, we also observed a substantial number of GluA1/A4 receptors that were not bound to NOE1, either because NOE1 dissociated during the purification or because it was not bound to begin with. To address the question of whether binding of NOE1 can restrict the conformational heterogeneity of native CP-AMPARs, we incubated purified GluA1-containing receptors with recombinant NOE1 (Fig. 4a). Following a 30-min incubation, we introduced anti-GluA4 7D8 Fab to label the untagged GluA4 subunits (Fig. 4b). A large cryo-EM dataset (Extended Data Fig. 8) revealed two predominant 2D classes. Approximately 75% of particles formed a NOE1–AMPAR complex with an ordered ATD layer and around 25% of complexes exhibited a disordered ATD layer (Fig. 4 and Extended Data Fig. 8b). Thus, recombinant NOE1 can bind to native CP-AMPARs and stabilize the ATD layer. Notably, most classes with a disordered ATD layer had at least one 11B8 scFv feature at the B or D position. This observation indicates that GluA1 is unable to establish a stable ATD dimer–dimer interface, which is consistent with the previously proposed model[5] and studies of homomeric GluA1 structures[25].

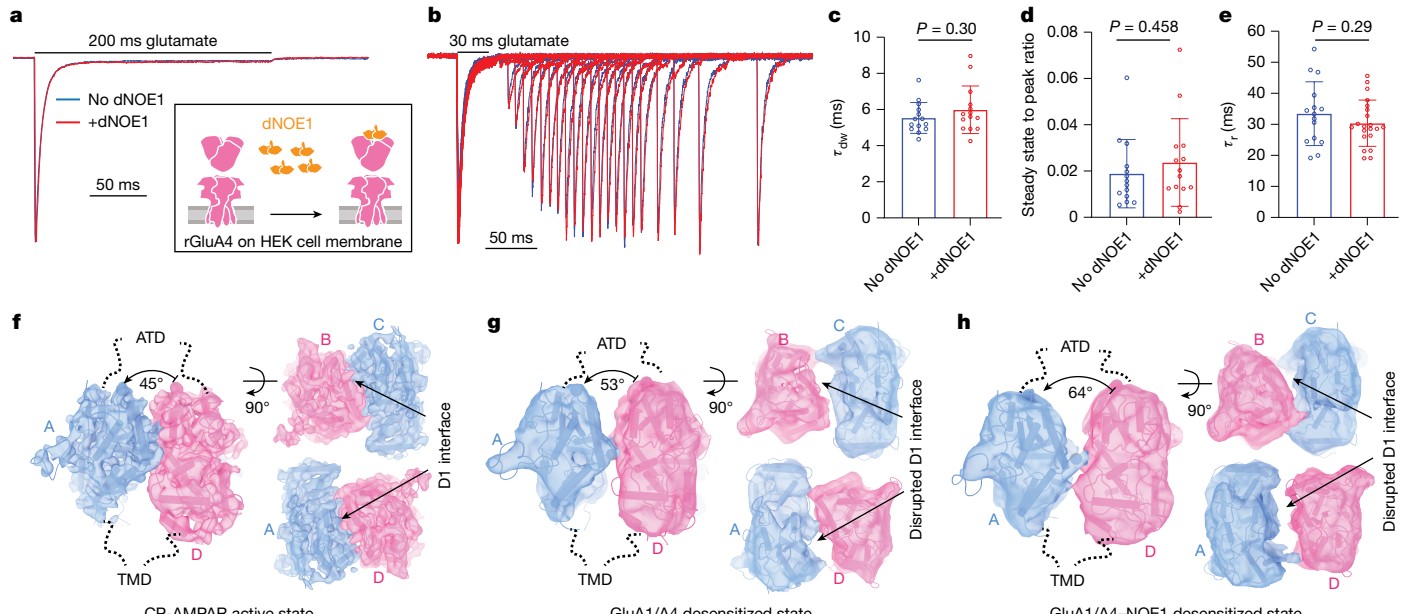

**Fig. 5 | NOE1 does not affect the desensitization kinetics of GluA4.**
**a**, Representative current responses to 10 mM glutamate applications were recorded from outside-out patches of HEK cells expressing AMPARs containing recombinant GluA4 (rGluA4) alone or incubated with 100 nM dNOE1.
**b**, Representative current responses show recovery from desensitization.
**c**–**e**, Desensitization kinetics of GluA4 with or without dNOE1. Statistical analyses were performed using two-sided $t$-tests. **c**, Dose-weighted desensitization time constants ($\tau_{dw}$) of GluA4 with ($n = 14$) or without dNOE1 ($n = 14$), calculated by weighting time constants according to their relative amplitude contributions.
**d**, The ratio of steady-state with the peak of GluA4 with ($n = 14$) or without dNOE1 ($n = 14$). **e**, Recovery time constants from desensitization ($\tau_r$) of GluA4 with ($n = 21$) or without dNOE1 ($n = 15$). $n$ denotes the number of recordings. **f**, The D1–D1 interface of cerebellar CP-AMPARs in the active state. **g,h**, The ruptured D1–D1 interface of cerebellar GluA1/A4 and GluA1/A4–NOE1 complexes in the desensitized state.

Further focused classification of the NOE1 ATD revealed a minor class (3%) with a single 7D8 Fab bound to the subunit at the A or C position, which we designate as an A1A4A4A4 subtype (Fig. 4b and Extended Data Fig. 8b–e). By contrast, most particles (approximately 72%) exhibited a subunit arrangement of A1A4A1A4 (Fig. 4b,e). These results suggest that the GluA1/A4 assembly is predominant in the cerebellum, with the GluA4 subunit preferentially occupying the B and D positions. This further shows that NOE1 is capable of binding to most CP-AMPARs in the cerebellum and is thus poised to have an important role in receptor localization and synaptic signalling.

Therefore, compared with GluA2-containing AMPARs, CI-AMPARs and CP-AMPARs share a similar overall architecture, comprising three distinct layers and similar auxiliary subunit occupancy. However, they differ in subunit stoichiometry and ATD conformation (Fig. 2 and Supplementary Fig. 4). In CI-AMPARs, GluA2 consistently occupies the B and/or D positions and is associated with a stable ATD layer (Supplementary Fig. 4), whereas GluA4 occupies the B and/or D positions in CP-AMPARs, with either stable or unstable ATD configuration.

## NOE1 does not affect desensitization

Previous studies have demonstrated that the ATD in GluA1 and GluA2 receptors modulates recovery from desensitization[25,49]. Given our finding that binding of NOE1 to GluA4 stabilizes the ATD layer (Fig. 4), we explored whether NOE1 influences the gating kinetics of GluA4-containing receptors. Moreover, our findings indicated that NOE1 binds to recombinant homomeric GluA4 receptors similarly to cerebellar CP-AMPARs, engaging with subunits at the B and D positions, and that GluA4 receptors can adopt either an ordered or disordered ATD layer when unbound with NOE1 (Supplementary Fig. 5).

We carried out rapid solution exchange, outside-out patch-clamp recordings of GluA4 receptors in the absence or presence of NOE1, inducing currents by the application of 10 mM glutamate (Fig. 5a,b).

We observed only small differences in the rate of desensitization, steady-state currents and recovery from desensitization that were not statistically significant (Fig. 5c–e), consistent with the small effect of NOE1 knockout on synaptic kinetics[43]. In contrast to previous studies on GluA1 and GluA2 receptors, constraining the ATD layer by way of NOE1 binding did not substantially affect these receptor gating kinetics. We speculate that because GluA4 receptors already exhibit fast recovery kinetics (approximately 33 ms) and approximately half of the population has a stable ATD layer at neutral pH, further stabilization of the ATD layer does not significantly accelerate recovery kinetics.

To further probe the gating mechanism of CP-AMPARs, we analysed the structures of CP-AMPARs in the activated state, in the presence of glutamate and positive allosteric modulator (PAM) ($R$, $R$)-2b, and in the desensitized states, with glutamate alone (Extended Data Figs. 9 and 10). In the active state, we obtained a map featuring four auxiliary subunits surrounding the TMD at a resolution of 4.0 Å, revealing a gating conformation similar to GluA2-containing receptors in the presence of positive allosteric modulator[3]. We identified a potentially open conductance path with kinked M3 helices at the B and D positions, whereas the M3 helices at the A and C positions exhibited only a subtle vertical shift (Extended Data Fig. 9d–f). Notably, recent studies indicate that GluA2-containing AMPARs associated with TARPs and CNIHs exhibit detectable $Ca^{2+}$ permeability[10]. Because these auxiliary subunits stabilize the M2 helices, they may induce conformational rearrangements that subtly reshape the pore and modify its electrostatic environment, which may explain how $Ca^{2+}$ permeation can occur even in RNA-edited, GluA2-containing AMPARs.

In the desensitized state (Extended Data Fig. 10), multiple rounds of heterogeneous refinement identified four major classes: two classes with ordered ATD layers and a twofold-symmetric LBD organization, specifically the subtypes A1A4A1A4 (13.4%) and A1A4A1A4 with NOE1 (22.3%); one class (35.4%) with a disordered ATD layer and a twofold-symmetric LBD layer; and another class (28.9%) with a

disordered ATD layer with LBDs arranged in a pseudo-fourfold symmetry, resembling desensitized kainate receptors or GluA2 without auxiliary proteins[50]. Refinement of these classes, focused on the LBD–TMD layer, yielded resolutions of 9.0 Å, 9.3 Å, 9.1 Å and 7.6 Å, respectively (Extended Data Fig. 10). The twofold-symmetric LBD layer maps of LBD–TMD reveal the presence of extracellular protrusions at the B′ and D′ positions, indicating the presence of auxiliary proteins. By contrast, the fourfold-symmetric LBD–TMD map does not exhibit any observable extracellular protrusions (Extended Data Fig. 10). A comparison of the desensitized states of the A1A4A1A4 and A1A4A1A4–NOE1 subtypes with the active state revealed a ruptured D1–D1 interface, similar to the desensitized state of the GluA2–TARP γ2 LBD conformation[4] (Fig. 5f–h). This observation indicates that A1A4A1A4 receptors maintain a conserved desensitized conformation, either bound or unbound to NOE1, suggesting that NOE1 does not influence the structural ensembles of the receptor in the desensitized state.

## NOE1 dimerizes receptor complexes

NOE1 has a substantial role in the lateral mobility of synaptic AMPARs[15] and is characterized by a V-shaped tetrameric structure[44]. By modelling the structure of intact, tetrameric NOE1 with AMPARs, we find that NOE1 has the potential to bind to two AMPARs. To validate this model, we conducted FSEC experiments with a low concentration of full-length NOE1–mScarlet and relatively high concentrations of GluA4 or CP-AMPARs (Extended Data Fig. 11 and Supplementary Fig. 8). The results show a peak around 2,250 kDa for the interaction of GluA4 with NOE1, and a peak around 2,685 kDa for the interaction of CP-AMPARs with NOE1 (Extended Data Fig. 11a,b), suggesting that these peaks correspond to one NOE1 tetramer bound to two receptors (Extended Data Fig. 11c). We speculate that NOE1 could function as a dimerization agent and—on interaction with additional synaptic proteins—may stabilize and organize AMPARs at the synaptic cleft[43], thereby enhancing AMPAR signalling. However, we note that further in vivo studies are required to establish the role of NOE1 in receptor dimerization or clustering.

## Discussion

We used dual immunoaffinity capture and selective protease release to isolate CP-AMPARs from the cerebellum. Single-particle cryo-EM analysis of these CP-AMPARs combined with biochemical and biophysical experiments showed that the major receptor population comprises GluA1 and GluA4 subunits arranged in an A1A4A1A4 assembly, demonstrating that the principle of non-stochastic subunit arrangement in AMPARs, first established with CI-AMPARs[5], also holds for CP-AMPARs. Analysis of the auxiliary subunit constellation defines receptor populations fully and partially occupied by TARPs. Those receptors that are fully occuped by TARPs are likely to represent plasma membrane receptors and those that are partially occupied demonstrate how the positioning of TARPs in AMPAR complexes is non-random.

The ATD layer of A1A4A1A4 receptors adopts two distinct conformations, one in which the layer is intact and the second in which the layer is conformationally heterogeneous. Among the group of particles with the ordered ATD layer, we identified a subset that was bound with NOE1, illustrating how NOE1 primarily interacts with the ATDs of GluA4 subunits at the B and D positions, stabilizing the ATD layer in a Y-shaped conformation. NOE1 binding to CP-AMPARs had no discernable effects on receptor desensitization kinetics. Thus, a synaptic scaffolding protein can restrain the conformational space of the ATD and LBD layers while having little effect on receptor function.

We further demonstrated how NOE1 can bind to two receptor complexes, thus providing insight into the mechanism underpinning NOE1 clustering of synaptic receptors and receptors at neuroglial appositions. We speculate that the interaction of the CP-AMPAR–NOE1

complex with additional synaptic proteins, such as neuritin and brorin[43] provides additional mechanisms for CP-AMPAR clustering and potentially AMPAR-mediated synaptic plasticity[7,15].

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

## Methods

### Expression and purification of anti-GluA1 Twin-Strep tagged 11B8 scFv–GFP

The previously described 11B8 scFv expression construct was modified by introducing a DNA sequence to connect the heavy and light chains of the variable domains with an (SGGGG)₃ linker and the resulting construct was cloned into the pFastBac1 vector[5]. The GP64 signal peptide was included at the N terminus and a green fluorescent protein (GFP) and a Twin-Strep II tag were introduced at the C terminus. The GFP and Twin-Strep tag were separated by a glycine-rich linker and a 3C protease site, using the following sequence GSGG–HRV 3C site–GSGS, yielding the 11B8 scFV–GFP–3C–Twin-Strep construct. Baculovirus derived from this construct was produced and SF9 insect cells, at a concentration of $2.0–3.0 \times 10^6$ cells per ml, were transduced with P2 virus and cultured at 20 °C for 96 h. Subsequently, the cells were pelleted by centrifugation at 6,400$g$ for 15 min, the supernatant was collected, and the pH was adjusted to 8.0 by adding Tris base powder. The supernatant was further clarified by centrifugation at 6,400$g$ for 20 min, passed through a 0.22-µm filter, and concentrated by tangential flow filtration using a 30-kDa molecular weight cutoff (MWCO) filter. When the medium reached a volume of approximately 200 ml, 1 l of TBS buffer (20 mM Tris, pH 8.0, 150 mM NaCl) was added to exchange the buffer and dilute the medium effectively. This process was repeated three times, stopping at a volume of approximately 200 ml. The 11B8 scFV was next purified by passage over and elution from Strep-Tactin resin and further purified by size-exclusion chromatography using a Superose 6 10/300 GL column pre-equilibrated with TBS. The resulting protein was stored in aliquots at −80 °C.

### Expression and purification of Twin-Strep tagged 15F1 Fab–mCherry

The 15F1 Fab expression construct was modified from a previous construct as follows[5]. The DNA sequences encoding the heavy and light chains of the Fab fragment were cloned into a bicistronic pFastBac1 vector. An mCherry fluorescent protein tag was introduced at the C terminus of the heavy chain and a Twin-Strep II tag was introduced at the C terminus of the light chain. A GS linker connected the Twin-Strep tag and the terminus of the light chain. The expression and purification of the 15F1 Fab–mCherry–Twin-Strep protein were the same as that for the 11B8 scFv.

### Purification of the anti-GluA3 5B2 monoclonal antibody and Fab

Purification of the GluA3-specific monoclonal antibody was carried out as described[5].

### Expression and purification of Twin-Strep tagged L21-32R Fab–mCherry

The anti-GluA2 C-terminal domain L21-32R monoclonal antibody was obtained from previous studies[51]. The DNA sequences encoding the heavy and light chains of the Fab fragment were cloned into a bicistronic pFastBac1 vector. An mCherry and Twin-Strep tag were introduced at the C terminus of the heavy chain. The GFP and Twin-Strep tag were separated by GSGG–HRV 3C site–GSGS. The expression and purification were the same as that for the 11B8 scFv construct.

### Generation and validation of 4H9 and 7D8 monoclonal antibodies and Fabs

Purified homomeric human GluA1 and rat GluA4 were separately reconstituted into proteoliposomes, and three mice per antigen were immunized for monoclonal antibody production as previously described[5]. Antibodies from isolated hybridoma cell clones were cross-screened using an ELISA with recombinant GluA1, GluA2, GluA3 or GluA4 receptors, and positive subunit-specific hits were further screened by FSEC. The top hit hybridomas were cultivated in a bioreactor, and antibodies were purified from hybridoma supernatant using Protein A/G resin. An Octet bio-layer interferometry system was used to measure the affinity of purified antibody hits with recombinant homomeric GluA antigens. The anti-GluA1 antibody 4H9 and anti-GluA4 antibody 7D8 were identified as low nanomolar affinity binders of their respective antigens, exhibiting undetectable off rates and high specificity across the GluA1–4 subunits at 40 nM. The 4H9 and 7D8 Fab fragments were produced by digesting the monoclonal antibody of 4H9 and 7D8 in the same manner as the GluA3 antibody 5B2, as described[5,40].

### Expression and purification of GluA1–4

Human GluA1 and rat GluA4 lacking the C-terminal tail and including a C-terminal Strep tag were cloned into the pEG-BacMam vector and used for monoclonal antibody production (Supplementary Table 1). Rat GluA1–4 and human GluA1–4, including a C-terminal GFP and Strep tag, were cloned into the pEG-BacMam vector and were used for monoclonal antibody screening and NOE1 specificity FSEC experiments. The bacmid and baculovirus were made using standard methods[52], and the following methods were applied to the expression and purification. P2 virus was added to tsA201 cells (ATCC CRL-11268) at 37 °C and 5% $CO_2$. At 12 h after infection, 10 mM sodium butyrate was added, and the temperature was changed to 30 °C. At 48 h after infection, the cells were collected by centrifuging at 6,400$g$ for 15 min. The pellet derived from 1 l of culture was resuspended in 25 ml TBS (20 mM Tris, pH 8.0, 150 mM NaCl) containing protease inhibitors at final concentrations of 1 mM phenylmethyl sulfonyl fluoride (PMSF), 0.8 µM aprotinin, 2 µg ml⁻¹ leupeptin, 2 µM pepstatin A. Digitonin was added to the mixture, at a final concentration of 2% (w/v), and the cells were solubilized for 90 min with slow stirring in the cold room. The resulting solution was clarified by ultracentrifugation (200,000$g$, 40 min, 4 °C), and the supernatant was collected and passed through Strep-Tactin resin by gravity three times in the cold room (ideally maintained at around 4 °C). After washing the column with 20–30 column volumes of SEC buffer (20 mM Tris, pH 8.0, 150 mM NaCl, and 0.075% digitonin), the bound proteins were eluted by 10 column volumes of FSEC buffer supplemented with 5 mM D-desthiobiotin. Receptor complexes were further purified by SEC using a Superose 6 10/300 GL column pre-equilibrated with SEC buffer. Peak fractions were pooled and concentrated using a micro concentrator fitted with a 100-kDa MWCO filter.

### Expression and purification of NOE1

For full-length NOE1 (Uniprot code: Q62609, residues 1–485), the DNA sequence encoding rat NOE1 was cloned into the pEG-BacMam vector. The Strep II or mScarlet-His tags were connected to the C terminus by an SG linker or a SG–HRV 3C site–AAAA motif. The bacmid and baculovirus were made using standard protocols[52], as further described below. P2 virus was added to tsA201 cells at 37 °C and 5% $CO_2$. At 12 h after infection, 10 mM sodium butyrate was added, and the temperature was changed to 30 °C. At 96 h after infection, the cells were centrifuged at 14,000$g$ for 15 min and the supernatant was collected. The supernatant was clarified by passage through a 0.22-µm filter and concentrated by tangential flow filtration using a 30-kDa MWCO filter. When the solution reached a volume of 200 ml, 1 l of TBSC buffer (20 mM Tris, pH 8.0, 150 mM NaCl, and 2 mM $CaCl_2$) was added to effect buffer exchange and dilute the cell culture media. This process was repeated three times, and the final solution was concentrated to a volume of approximately 200 ml. NOE1 was isolated by Strep-Tactin (NOE1–Strep) or Ni-NTA (NOE1–mScarlet-His) affinity chromatography. Proteins were further purified by SEC using a Superose 6 10/300 GL column pre-equilibrated with TBSC and stored at −80 °C.

To make the construct for the dimer of NOE1 (dNOE1), the DNA sequence encoding NOE1 residues spanning 205 to 481 was cloned into the pEG-BacMam vector. The cystatin SN signal peptide and five residues from the coiled-coil region of GCN4 (MAQYLSTLLLLLAT-LAVALAGLKQEI)[53] were included at the N terminus and an mScarlet

fluorophore, together with a polyhistidine tag, was introduced into the construct at the C terminus and connected via a GSG–HRV 3C site–AAAA linker. For the dNOE1 pull-down assay, the Twin-Strep tag and HRV 3C site were inserted immediately following the signal peptide. The expression and purification of dNOE1 followed the same protocol as full-length NOE1, except that it was not concentrated by tangential flow filtration prior to isolation using Ni-NTA.

### Purification of GluA1-containing CP-AMPARs from cerebellum

To isolate cerebellar GluA1-containing CP-AMPARs, we used donated cerebellum tissue derived from male and female *Rattus norvegicus* between the ages of 6–12 weeks. The cerebellums of 15 rats were placed in 10 ml of TBS buffer containing protease inhibitors at final concentrations of 1 mM PMSF, 0.8 µM aprotinin, 2 µg ml$^{-1}$ leupeptin, and 2 µM pepstatin A, and were homogenized by a Teflon-glass blender and further disrupted by sonication on ice, using a sonicator equipped with a microtip, for 90 s with 3 s on and 5 s off, typically at a 20 W output. Before sonication, 50 µM (*R*,*R*)-2b and 2 µM MPQX were added to the solution, to stabilize the CP-AMPARs in closed, non-desensitized conformations. Next, an additional 10 ml of TBS buffer was added to the crude mixture, and the membranes were solubilized in 2% (w/v) digitonin for 30 min with slow stirring in the cold room. The resulting solution was clarified by ultracentrifugation at 200,000g and the supernatant was collected and mixed with excess 11B8 scFV–GFP–3C–Twin-Strep and Twin-Strep-tagged 15F1 Fab–mCherry. Subsequently, the solution was passed through Strep-Tactin XT resin by gravity three times. The column was then washed with 20–30 column volumes of FSEC buffer (20 mM Tris, pH 8.0, 150 mM NaCl, 0.075% digitonin, 10 µM (*R*,*R*)-2b and 2 µM MPQX). Then, bound proteins were eluted with 1 column volume of 0.5 mg ml$^{-1}$ 3C protease in FSEC buffer after 10 min incubation in the cold room. The resin was washed with two column volumes of FSEC buffer. The eluted sample was incubated with an excess of 5B2 Fab at 4 °C before injecting it into a Superose 6 10/300 GL column equilibrated in FSEC buffer. Peak fractions were pooled and concentrated using a 100-kDa MWCO concentrator to an OD$_{280}$ of 0.12 mg ml$^{-1}$. The total time from solubilization to final isolation was approximately 3.5 h. The yield of the receptor was around 4 µg, sufficient for subsequent cryo-EM analysis using 2 nm carbon support grids.

### Purification of GluA2-containing AMPARs from whole brain

The procedure for isolating native GluA2-containing AMPARs is the same as for CP-AMPARs. Ten whole mouse brains were homogenized and solubilized. The supernatant was collected and mixed with excess L21-32R Fab–mCherry–3C–Twin-Strep II, for 10 min at 4 °C. Gravity passed the solution through a column packed with Strep-Tactin XT resin. The bound proteins were eluted by 1 column volume of 0.5 mg ml$^{-1}$ 3C protease in FSEC buffer after 10 min incubation in the cold room. Then the sample was incubated with an excess of 11B8 scFv and concentrated. Proteins were further purified by FSEC and peak fractions were pooled and concentrated using a 100-kDa MWCO concentrator.

### dNOE1 pull-down from rat cerebellum

The procedure for the dNOE1 pull-down experiment is the same as the 11B8 scFv pull-down. The solubilized cerebellum supernatant was collected and mixed with excess Twin-strep–3C–dNOE1–mScarlet protein (final ~70 nM), and the mixture was then passed through Strep-Tactin XT resin by gravity three times in a cold room. After washing the column with 20–30 column volumes of FSEC buffer (20 mM Tris, pH 8.0, 150 mM NaCl, and 0.075% digitonin, 10 µM (*R*, *R*)-2b, 2 µM MPQX) the bound proteins were eluted by 1 column volume of 0.5 mg ml$^{-1}$ 3C protease after 10 min incubation in a cold room. The resin was washed with 2 column volumes of FSEC buffer. The target receptor complexes were further purified by FSEC and peak fractions were pooled and concentrated using a 100-kDa MWCO concentrator to an OD$_{280}$ of 0.07 mg ml$^{-1}$.

### SiMPull assay

Rat cerebellum tissue was homogenized with a Dounce homogenizer in homogenization buffer (20 mM Tris-HCl pH 8.0, 150 mM NaCl, 0.8 µM aprotinin, 2 µg ml$^{-1}$ leupeptin, 2 µM pepstatin A) in 7.5 ml buffer per gram of tissue. The homogenized sample was diluted 1:1 with solubilization buffer (20 mM Tris-HCl pH 8.0, 150 mM NaCl, 0.8 µM aprotinin, 2 µg ml$^{-1}$ leupeptin, 2 µM pepstatin A, 4% w/v digitonin) and gently mixed at 4 °C for 15 min. The solubilized sample was spun at 4,000g for 3 min at 4 °C and the supernatant was collected and passed through a 0.22-µm filter. SiMPull experiments were conducted in SiMPull assay buffer (20 mM Tris-HCl pH 8.0, 150 mM NaCl, 0.075% w/v digitonin, 0.2 mg ml$^{-1}$ BSA). Passivated and biotinylated TIRF slides with flow chambers were prepared as previously described[40]. To each chamber, 10 µl of 0.25 mg ml$^{-1}$ streptavidin was applied and incubated for 5 min. Chambers were washed with 30 µl of assay buffer, then 10 µl of the anti-GluA1 capture antibody conjugated to biotin (4H9mAb–biotin) was applied at 20 nM concentration and the slide was incubated for 5–10 min. A control without capturing antibody applied to the slide was included as a negative control for each experiment. Chambers were washed with 30 µl of assay buffer, and 30 µl of rat cerebellum solubilized supernatant diluted 1:50 in assay buffer was applied and incubated for 5–10 min. Chambers were washed with 30 µL of assay buffer, and detection antibodies for GluA1 (11B8mAb–Alexa Fluor 488), GluA2 (15F1mAb–Alexa Fluor 555) and GluA4 (7D8mAb–Janelia Fluor 646) were applied at 200 nM and incubated for 20 min. Chambers were washed with 30 µL of assay buffer and imaged on a Leica DMi8 TIRF microscope with an oil-immersion 100× objective using an Andor iXon Ultra 888 back-illuminated EMCCD camera with a 133 × 133 µm imaging area and a 130 nm pixel size, following a previously described protocol[40]. For counting the fluorophore spots and colocalization, different fluorophore images were acquired in the same region under different wavelengths and the position of each molecule was calculated in the centre area (18 × 18.5 µm) of the image using ComDet in Image J (FIJI). The spot intensity threshold was set at 5. Molecules located within a six-pixel radius were colocalized.

For the purified CP-AMPARs, the SiMPull experiments were carried out by applying the sample to imaging chambers coated with the GFP nanobody to capture 11B8 scFv–GFP. Biotinylated anti-GFP nanobody at 10 µg ml$^{-1}$ was applied to the slide, incubated for 10 min, and unbound material was washed off with 30 µl of buffer A (20 mM Tris, pH 8.0, 150 mM NaCl, 0.075% (w/v) digitonin, 2 µM MPQX, and 0.25 mg ml$^{-1}$ BSA). The CP-AMPARs were applied to the chamber at a concentration of approximately 100 pM to visualize GluA2-3-containing complexes and at approximately 50 pM to visualize GluA4-containing complexes. The chamber was incubated for 10 min, followed by washing with 30 µl of buffer A. Alexa Fluor 647-labelled detection antibodies were applied to the chamber at a concentration of 10 µg ml$^{-1}$ for 10 min, followed by washing with 30 µl of buffer A. The chamber was then immediately imaged using the TIRF microscope. To estimate nonspecific binding, we did not apply the biotinylated anti-GFP nanobody to the slide, yet the other steps were the same as in the experimental chamber. The spot intensity threshold was set up for the GFP fluorescence spots at 3.6. Molecules located within a six-pixel radius were colocalized.

For the full-length NOE1 binding specificity assay, the recombinant AMPARs were applied to the chamber at a concentration of 30 pM and captured by anti-GFP nanobody for 10 min, followed by washing with 30 µl of buffer A. The full-length NOE1–mScarlet was applied to the chamber at a concentration of 20 nM and incubated for 10 min, followed by washing with 30 µl of buffer A and subsequent imaging. To estimate nonspecific binding, we did not apply the biotinylated anti-GFP nanobody to the slide, while keeping the other steps the same as in the experimental chamber. The GFP and mScarlet fluorescence spot intensity threshold was set at 3.6. Molecules located within a six-pixel radius were colocalized. Each SiMPull experiment was repeated over two independent experiments.

## Cryo-EM sample preparation and data acquisition

For the GluA1-containing CP-AMPARs in the closed state, 3 µl of 0.12 mg ml$^{-1}$ CP-AMPAR was applied to Quantifoil 2/1 200 mesh gold grids layer with 2 nm continuous carbon, which were glow-discharged in the presence of amyl amine for 30 s at 15 mA. After 15 s of waiting, the grids were blotted for 3 s with a blot force of 0 and flash-frozen in liquid ethane using a Vitrobot at 16 °C and 100% humidity.

For the GluA1-containing CP-AMPARs in the desensitization state, 3 µl of CP-AMPAR was quickly added and pipetted up and down in a mixture of 0.3 µl 200 mM glutamate, with a final concentration of 20 mM glutamate, and the solution was quickly applied to Quantifoil 2/1 200 mesh gold grids covered with 2 nm of continuous carbon, which were glow-discharged, blotted and flash-frozen as described above.

Cryo-EM data were collected using a Krios microscope fitted with a Falcon 4 detector at a physical pixel size of 0.94 Å with an energy filter set to a 20-eV slit width and a defocus range of −1.2 to −2.2 µm. Images were collected using 'multi-shot' and 'multi-hole' methods, permitting the acquisition of 8 images per hole, from 9 neighbouring holes (3 × 3) per stage shift. Each image was collected at an exposure rate of around 8.0 e$^{-}$ per pixel per second, for a total exposure time of 5.21 s, resulting in a total dose of 50 e$^{-}$ Å$^{-2}$.

For the GluA1-containing CP-AMPARs in the active state, 0.15 mg ml$^{-1}$ CP-AMPAR was incubated for 30 min with 50 µM ($R$, $R$)-2b, then 3 µl of this was quickly added and pipetted up-and-down in a mixture of 0.3 µl 200 mM glutamate (final concentration 20 mM). The cryo-EM sample preparation was performed as described above. Cryo-EM data were collected using a Krios microscope fitted with a Gatan K3 detector at a pixel size of 0.4155 Å in super-resolution mode with an energy filter set to a 20-eV slit width, and a defocus range of −1.2 to −2.2 µm. Each image was collected with 50 frames at an exposure rate of around 8.0 e$^{-}$ per pixel per second, for a total exposure time of 4.26 s, resulting in a total dose of 50 e$^{-}$ Å$^{-2}$.

For the GluA1-containing CP-AMPARs with recombinant NOE1, the purified CP-AMPAR was incubated with excess NOE1–Strep (final ~500 nM) for 30 min on ice, excess 7D8 Fab and 5B2 Fab were added, and the resulting complex was purified by FSEC using a Superose 6 10/300 GL column equilibrated in FSEC buffer (20 mM Tris, pH 8.0, 150 mM NaCl, and 0.075% digitonin, 10 µM ($R$, $R$)-2b, 2 µM MPQX). Peak fractions were pooled and concentrated using a 100-kDa MWCO concentrator to an OD$_{280}$ of 0.1 mg ml$^{-1}$. The cryo-EM sample preparation was performed as described above. Cryo-EM data were collected using a Krios microscope fitted with a Gatan K3 detector at a pixel size of 0.53 Å in super-resolution mode with an energy filter set to a 20-eV slit width, and a defocus range of −1.2 to −2.2 µm. Each image was collected at an exposure rate of around 25 e$^{-}$ per pixel per second, for an exposure time of 2.5 s, resulting in a total dose of 55.7 e$^{-}$ Å$^{-2}$.

For recombinant GluA4 with NOE1, the GluA4 receptor was incubated with NOE1 in a molar ratio 5:1, respectively, for 30 min in a Tris buffer (20 mM Tris, pH 8.0, 150 mM NaCl, and 0.075% digitonin, 50 µM ($R$, $R$)-2b, 1 µM MPQX). A 3 µl volume of the 3 mg ml$^{-1}$ mixture was applied to Quantifoil 1.2/1.3 300 mesh gold holey carbon grids, which were glow-discharged for 30 s at 15 mA. The grids were blotted and flash-frozen as described above. Cryo-EM data were collected via a Krios microscope configured with a Falcon 4 detector at a pixel size of 1.196 Å in super-resolution mode with an energy filter set to a 20-eV slit width, and a defocus range of −1.8 to −2.5 µm. Each image was collected at an exposure rate of around 10.3 e$^{-}$ per pixel per second, for a total exposure time of 8.6 s, resulting in a total dose of 62 e$^{-}$ Å$^{-2}$.

## Cryo-EM data processing

**Motion correction and CTF estimation.** All cryo-EM images were processed by cryoSPARC (v4.1-4.6)[54]. Beam-induced motion was corrected by patch motion correction, and contrast transfer function (CTF) parameters were determined by patch CTF estimation.

## GluA1-containing CP-AMPARs in the closed state

A random set of 4,618 images was selected and particles were picked using the Blob picker in cryoSPARC to generate 2D class averages with clear receptor features, which were used for template picking. The non-clarified blob-picked particles were reconstructed by ab initio reconstruction ($n$ = 3) as bad references. Particles from the entire dataset (29,662 movies) were then picked using template-based picking and Blob picking, and particles were extracted with a box size of 448 × 448 pixels and down-sampled by 4× binning. Several rounds of 2D classification were carried out and only classes showing clear AMPAR features were kept, and duplicate particles were removed, resulting in the retention of 575,426 particles. The particles were then reconstructed using ab initio reconstruction ($n$ = 6) as good references. Two rounds of heterogeneous refinement were employed to remove 'junk' particles from the whole dataset, resulting in the retention of 552,695 particles, 279,832 of which were of a noelin-containing AMPAR complex, showing clear noelin binding at the B and D positions and two 11B8 scFvs at the A and C positions (class 2 in Fig. 2). Combining 2D classification (575,426) and heterogeneous refinement (552,695) resulted in 782,253 particles after removing duplicates. The particles were then re-extracted to a box size of 448 × 448 pixels (bin 1). After two rounds of ab initio and heterogeneous refinement, 492,149 particles were kept, giving rise to 5 different populations. Two of them have a blurry ATD layer and no clear antibody features (class 5 and 6 in Fig. 2). One class has a single 11B8 scFv binding on the A/C subunit, but another ATD dimer is blurry (class 4 in Fig. 2). Another class has a splayed ATD layer with a single scFv at the B or D position, and another ATD dimer is also blurry (class 3 in Fig. 2). One class containing 147,476 particles showed clear 11B8 scFv features binding at the A and C subunits, but no extra density features at the B and D subunits, suggesting that this class is a GluA1/A4 heterotetramer, where GluA1 occupies the A and C positions and GluA4 occupies the B and D positions (class 1 in Fig. 2). Through heterogeneous refinement of this class, we identified two distinct subclasses: one with a stable ATD layer and another with a flexible ATD layer. Further non-uniform refinement of the stable ATD subclass yielded the following results: a 4.29 Å overall map (C1, bin 1) for the GluA1/A4 receptor and a 3.53 Å ATD layer map (C2, bin 1) for GluA1/A4, by focusing on the stable ATD layer, subtracting the LBD–TMD layer. Further 3D classification focused on the TMD layer, subtracting the ATD layer and without particle alignment, yielded a 4.18 Å LBD–TMD layer map (C2, bin 2) for GluA1/A4 with four auxiliary proteins positioned at A', B', C' and D', together with a 4.22 Å LBD–TMD layer map (C2, bin 2) for GluA1/A4 with two auxiliary proteins localized at B' and D', by non-uniform refinement.

To identify the auxiliary proteins of the LBD–TMD layer, we combined all classes except the noelin-containing complex and used focused classification of the LBD–TMD layer. First, the ATD layer was subtracted from each class, and two rounds of heterogeneous refinement ($n$ = 4) were performed. The resulting 163,159 particles showed clear LBD–TMD features with clear auxiliary protein densities at the B' and D' positions, with prominent extracellular protrusions, but also showed weak densities at the A' and C' positions. We therefore performed focused 3D classification (i, $n$ = 3) using a soft mask around the A' and C' position and the TMD, resulting in one class (class 3) that displayed weak density at the A' and C' positions. One more round of 3D classification ($n$ = 2) was performed, still focusing on the A' and C' positions, resulting in one class containing 30,862 particles displaying continuous transmembrane helical densities at the A' and C' positions. However, when the threshold of the map was contoured to a lower value, we visualized weak densities for extracellular protrusions at the A' and C' positions. Therefore, one more round of 3D classification ($n$ = 2) was performed, focusing on the extracellular protrusions of the A' and C' position, resulting in two classes that contained 16,060 and 14,802 particles, respectively. Non-uniform refinement for these two classes

yielded a 3.99 Å LBD–TMD map (C2, bin 1) with four auxiliary proteins at A'–D' which all show prominent extracellular protrusions, and a 4.03 Å LBD–TMD map (C2, bin 1) also with four auxiliary proteins at A', B', C' and D' positions, but the density of A' and C' were weak and without extracellular protrusions. For the other two classes of 3D classification (i), we performed a round of 3D classification that focused on the B' and D' positions and TMD, yielding a 56,662-particle stack and a 3.74 Å LBD–TMD map (C2, bin 1) that only displayed density at the B' and D' positions with prominent extracellular protrusions and with no discernable density features at the A' and C' positions.

For the noelin-containing AMPARs complex, two rounds of heterogeneous refinement were performed for 279,832 particles, resulting in 208,802 particles. To obtain a good map for the noelin–ATD complex, we subtracted the LBD–TMD layer and performed heterogeneous refinement. The best class, consisting of 143,111 particles, was then subjected to non-uniform refinement, resulting in a 3.26 Å map (C2, bin 1) by the gold-standard Fourier shell correlation (FSC) (0.143). To obtain the optimal map for the LBD–TMD layer, we subtracted the ATD layer and performed heterogeneous refinement. The best class, consisting of 103,981 particles, was then subjected to two rounds of 3D classification with a soft mask around TMD and auxiliary subunits, resulting in 24,345 particles showing clear density for the A'–D' positions and continuous TMD transmembrane helixes. A total of 24,345 particles were then subjected to non-uniform refinement, resulting in a 4.2 Å map (C2, bin 2) by the gold-standard FSC (0.143). This map shows four auxiliary subunit densities around the TMD at A', B', C' and D', where the B' and D' positions show prominent extracellular protrusions, but A' and C' do not.

## GluA1-containing CP-AMPARs in the desensitized state

A random set of 4,489 images was selected and particles were picked using the Blob picker to generate 2D class averages with clear receptor features, which then were used for template picking. The non-clarified blob-picked particles were reconstructed by ab initio reconstruction (*n* = 2) as bad references. The entire dataset (16,939 images) was then picked using template-based picking and blob picking, and particles were extracted with a box size of 480 × 480 pixels and down-sampled by 4× binning. Several rounds of 2D classification were carried out and only classes showing clear AMPAR features were kept. The particles were then reconstructed using ab initio reconstruction (*n* = 5) as good references. Four rounds of heterogeneous refinement were used to clarify particles in the whole dataset, resulting in the retention of 950,851 particles. One class showed noelin binding with the AMPARs, containing 219,522 particles. Further heterogeneous refinement (*n* = 3) for this class yielded two classes that displayed canonical receptor features. One class containing 73,648 particles showed a well-resolved ATD layer with 11B8 scFvs at the A and C positions, but without noelin, suggesting that this is a GluA1/A4 receptor. Following further focused heterogeneous refinement, subtracting the ATD layer, the best class consisting of 28,959 particles was then subjected to non-uniform refinement, resulting in a 9.02 Å map (C1, bin 2) by the gold-standard FSC (0.143). Another class contained 122,391 particles showing noelin binding at the B and D positions and two 11B8 scFvs at the A and C positions. After further focused heterogeneous refinement and subtraction of the LBD–TMD layer, the best class consisting of 22,317 particles was then subjected to non-uniform refinement, resulting in a 9.34 Å map (C1, bin 2) by the gold-standard FSC (0.143). Because functional desensitized states exhibit considerable, unresolvable conformational heterogeneity, the resolution of corresponding structural reconstructions is limited. Moreover, there are a relatively small number of particles, which further limits the resolution of the reconstructions.

## GluA1-containing CP-AMPARs with recombinant NOE1

A random set of 7,747 images was selected and particles were picked using the Blob picker in cryoSPARC to generate 2D class averages with

clear receptor features, which then were used for template picking. The entire dataset was next picked using template-based picking and Blob picking, and particles were extracted with a box size of 480 × 480 pixels and down-sampled by 4× binning. Several rounds of 2D classification were carried out and only classes showing clear AMPAR features were kept, and duplicate particles were removed, resulting in the retention of 639,776 particles. The particles were then re-extracted to a box size of 480 × 480 pixels (bin 1), and one round of 2D classification was performed. The classes showing clear AMPAR features were kept, resulting in the retention of 511,593 particles. One additional round of 2D classification was performed, and the resulting classes were analysed. Particles showing NOE1-containing AMPAR complexes were selected, resulting in a particle stack of 376,106 particles (75.1%). Additionally, 124,413 particles (24.9%) showed an unstable ATD layer and at least one 11B8 scFv at the B or D position, indicating that at least one GluA1 subunit was present at the B or D position in this class. For the NOE1–AMPAR complex, particles were subjected to non-uniform refinement, followed by subtraction of the LBD–TMD layer. To determine whether this class contained only one 11B8 scFv, a 3D classification (*n* = 5) was performed without particle alignment. The results revealed a class containing 81,731 particles that displayed only a single 11B8 scFv signal. At a lower threshold, weak density resembling a Fab was observed. Subsequently, two additional rounds of 3D classification were performed using a soft mask around the ATD monomer and the Fab. Among the resulting classes, one containing 15,071 particles showed a single 11B8 scFv and a Fab bound to the A and C positions of the ATDs. This class was subjected to non-uniform refinement, yielding a 4.32 Å map (C1, bin 1) as determined by the gold-standard FSC (0.143). Based on the binding features of 5B2, 11B8, 15F1 and 7D8, the Fab observed in this map was identified as the anti-GluA4 7D8 Fab. Therefore, this class is A1A4A4A4. The remaining particles, which contained only a single 11B8 scFv, underwent non-uniform refinement, resulting in a 3.61 Å map (C1, bin 1) by the gold-standard FSC (0.143). However, structural details of this class—such as glycosylation (N45 and N239, Extended Data Fig. 8)—confirmed this no antibody fragment-bound subunit at the A or C position was GluA1, suggesting that the 11B8 scFv had dissociated.

## GluA1-containing CP-AMPARs in the active state

A random set of 6,679 movies was selected and particles were picked using the Blob picker to generate 2D class averages with clear receptor features, which then were used for template picking. The non-clarified template-picked particles were reconstructed by ab initio reconstruction (*n* = 3) as bad references. The entire dataset (13,584 movies) was then picked using template-based picking and blob picking, and particles were extracted with a box size of 512 × 512 pixels and down-sampled by 4× binning. Several rounds of 2D classification were carried out and only classes showing clear AMPAR features were kept. The particles were then reconstructed using ab initio reconstruction (*n* = 3) as good references. Three rounds of heterogeneous refinement were used to clarify particles in the whole dataset, resulting in the retention of 778,925 particles. One more round of ab initio reconstruction and heterogeneous refinement (*n* = 8) yielded a retention of 642,084 particles. Combining 2D classification (608,996) and heterogeneous refinement (642,084) resulted in 858,480 particles after removing duplicates. The particles were then re-extracted to a box size of 512 × 512 pixels (bin 1). Further ab initio reconstruction and heterogeneous refinement (*n* = 8) were performed. The class showed clear features of noelin-bound receptors and was subjected to two more rounds of heterogeneous refinement, resulting in 27,260 particles without noelin bound. All receptor particles without noelin bound were put through homogeneous refinement and subtraction of the ATD signals, and the box size was reduced to 360 × 360 pixels (bin 1). Then two rounds of heterogeneous refinement were performed, resulting in 154,899 particles showing clear LBD–TMD features. One round of 3D classification (*n* = 2) without particle alignment with a soft mask around the TMD and the B' and

D' positions was performed, resulting in 78,324 particles displaying continuous transmembrane helical densities at the B' and D' positions, with prominent extracellular protrusions. Further, 3D classification ($n$ = 3) without particle alignment with a soft mask around TMD and the A' and C' positions resulted in 26,641 particles displaying clear densities at the A' and C' positions. The particles were then subjected to non-uniform refinement, resulting in a 3.96 Å map (C2, bin 1) by the gold-standard FSC (0.143). The map is shown in an open state and, as an indication of the quality of the map, shows clear ($R,R$)-2b signals.

### Recombinant GluA4 with NOE1

Blob picker was used to pick particles in cryoSPARC to generate 2D class averages with clear receptor features, which were used for template picking. After template-based picking, particles were extracted with a box size of 416 × 416 pixels and down-sampled by 4× binning. Several rounds of 2D classification were carried out and only classes showing clear AMPAR features were kept. The particles were then reconstructed using ab initio reconstruction ($n$ = 4) as references. Three rounds of heterogeneous refinement were performed to clarify the particles. In one class that showed NOE1 binding with the AMPARs, containing 34,395 particles, the particles were then re-extracted to bin 1 and subjected to non-uniform refinement, resulting in a 7.44 Å map (C1, bin 1) by the gold-standard FSC (0.143). One class showed a stable ATD layer, containing 44,360 particles, and the particles were then re-extracted at bin 1 and subjected to non-uniform refinement, resulting in an 8.02 Å map (C1, bin 1) by the gold-standard FSC (0.143). For one class that showed a blurry ATD layer, containing 54,376 particles, the particles were re-extracted to bin 1 and subjected to non-uniform refinement, resulting in a 4.61 Å map (C1, bin 1) by the gold-standard FSC (0.143).

### Model building

The structural modelling of the A1A4A1A4–scFv complex was carried out using rigid body fitting of the structure of A1A2A1A2 (PDB ID: 7LDD) and AlphaFold2 predicted models from the Alpha Fold DB using UCSF Chimera[5,55,56]. The structure was manually adjusted in Coot, with stereochemical restraints applied[57] and further refined by real-space refinement using Phenix[58]. All other structures were modified from the A1A4A1A4–scFv complex and manually adjusted in Coot and refined in real space using Phenix.

### Outside-out patch recording

A density of $2.5 \times 10^6$ HEK293S GnTi$^-$ 15 cells[59] were infected with baculovirus carrying the cDNA encoding GFP-tagged full-length GluA4 at a multiplicity of infection (MOI) of approximately 2. Cells were incubated at 37 °C with 8% $CO_2$ for 10 h. Subsequently, about $0.5 \times 10^6$ cells were plated onto 12 mm poly-L-lysine coated glass coverslips (Fisher Scientific, Cat#08774383) in a 35 mm dash and the competitive antagonist (NBQX, 30 μM) was added to cells. The cells were incubated at 30 °C and 8% $CO_2$ in DMEM (Corning, Cat#10-013-CV) containing 10% FBS. Recordings were done from 26 h–36 h post-infection.

Fast perfusion experiments were performed with a two-barrel theta tube. The tip of the theta tube was cut to around 300 μm, and etched with 10% hydrofluoric acid (in ethanol) for 25 min. The theta tube was mounted on a piezoelectric translator and moved by applying voltage (10 V) that was filtered with a 40 Hz Bessel filter to reduce mechanical oscillations of the piezoelectric device. At the end of recordings, junction currents evoked by changes in open-tip potentials were measured. The 20–80% rise times of these currents gave solution exchange times of approximately 700 μs. All experiments were performed at room temperature. The holding potential was −60 mV with a pipette solution containing (in mM): 135 CsF, 10 CsCl, 10 EGTA, 10 HEPES, 2 Na$_2$-ATP, and 0.1 spermine, adjusted to pH 7.3 with CsOH. The external solution was composed of (in mM) 145 NaCl, 3 KCl, 2 CaCl$_2$, 1 MgCl$_2$, 10 glucose, and 10 HEPES, adjusted to pH 7.4 using NaOH. The currents were elicited by 10 mM L-glutamate. To investigate the kinetics of receptor gating in the presence of NOE1, a solution containing 100 nM of dNOE1 was added to the perfusion system and bath solution. Before patching, the cells were incubated in the bath solution for 5–10 min. Recovery from desensitization was obtained from the two-pulse protocol. A conditioning pulse of 10 mM glutamate with a duration of 30 ms was followed by 10 ms glutamate pulses delivered at intervals increasing by 8 ms. Data were acquired with the Axoclamp 200B and analysed with Clampfit, and GraphPad Prism.

### $^3$H-AMPA binding assay

Wheat germ agglutinin-coated YSi SPA beads were used for the scintillation proximity assay. The concentrations of CP-AMPAR and $^3$H-AMPA used in the experiments were 20 nM and 100 nM, respectively. The experiments were recorded in counts per minute for approximately 12 h at room temperature. Nonspecific binding was determined by including 10 mM glutamate in the binding reaction mixtures. Three parallel experiments were conducted, and the mean values and standard errors were calculated.

### FSEC experiments

All FSEC experiments were performed using a UFLC instrument (Shimadzu) with a Superose 6 10/300 GL column operating at a flow rate of 0.5 ml min$^{-1}$. For FSEC experiments of dNOE1 with GluA1–4, 5 nM dNOE1–mScarlet was incubated with 200 nM GluA1, A2, A4, A3 or 15–150 nM GluA4, respectively, in FSEC buffer (20 mM Tris, pH 8.0, 150 mM NaCl, 2 μM ZK, 10 μM ($R,R$)-2b, and 0.075% digitonin) for 10 min. For FSEC peak shift experiments of dNOE1 with GluA4 mutants, 50 or 200 nM dNOE1–mScarlet was incubated with 10 nM GluA4 in FSEC buffer for 10 min. The mixture (70 μl) was loaded onto the Superose 6 10/300 GL column equilibrated with FSEC buffer. The eluent was passed through a fluorometer with settings for excitation of 561 nm and emission of 610 nm, with a recording time of 55 min. For FSEC experiments for full-length NOE1 with high concentrations of GluA4 or CP-AMPARs, 5 nM full-length NOE1–mScarlet was incubated with 5–150 nM GluA4, or 1 nM full-length NOE1–mScarlet was incubated with 1–100 nM CP-AMPARs in FSEC buffer (20 mM Tris, pH 8.0, 150 mM NaCl, 2 μM ZK, 10 μM ($R,R$)-2b, and 0.075% digitonin) for 10 min. The mixture (70 μl) was loaded onto the Superose 6 10/300 GL column equilibrated with FSEC buffer. The eluent was detected by a fluorometer as described above. Calibration of approximate molecular weight relied on a standard protein curve derived from standard molecular weight markers (Sigma, Cat# MWGF1000): 70 μl of 1.5 mg ml$^{-1}$ thyroglobulin, 0.5 mg ml$^{-1}$ β-amylase, 1 mg ml$^{-1}$ albumin, or 0.5 mg ml$^{-1}$ carbonic anhydrase.

### Measurement of dNOE1 binding affinity by Octet bio-layer interferometry

To measure the affinity between dNOE1 and GluA4, we made a construct that contained a fusion of a mouse Fc fragment before the dNOE1 coiled-coil domain. The expression and purification of the Fc fusion construct were the same as for dNOE1–mScarlet. This assay was performed by Octet RED384 at 30 °C. The Fc–dNOE1–mScarlet, at 20 μg ml$^{-1}$, was captured by the AMC biosensor for 10 min, the baseline was allowed to stabilize in buffer A (20 mM Tris, pH 8.0, 150 mM NaCl, and 0.075% digitonin with 3 mM glutamate or 2 μM ZK and 10 μM ($R,R$)-2b) for 5 min, and then progressively higher concentrations of GluA4–GFP in buffer A (0.78, 1.56, 3.13, 6.25, 12.5, 25 or 50 nM) were added over 15 min. Measurement was carried out for 30 min in buffer A. Raw data were pre-processed, analysed and fitted using the 1:1 binding model in the Octet data analysis software (10.0, Sartorius) to generate kinetic parameters.

### Western blot analysis

Purified cerebellum CP-AMPARs were loaded onto SDS–PAGE gels and transferred to a nitrocellulose membrane. Antibodies used for

detection were anti-GluA1 (Millipore 04-823, 1:1,000), anti-GluA2 (Thermo Fisher PA5-19496, 1:1,000), anti-GluA3 (Invitrogen 32-0400, 1:1,000) and anti-GluA4 (Millipore ab1508, 1:1,000), IRDye 800 CW anti-mouse/rabbit secondary antibodies were used for visualization. Blots were developed by adding secondary antibodies at a ratio of 1:10,000.

## Animal use statement

Rat carcasses were donated from other laboratories of the Vollum Insitute. No randomization, blinding or experimental manipulations were performed on these animals. All rats were euthanized under the OHSU Institutional Animal Care and Use Committee (IACUC) protocols, consistent with the recommendations of the Panel on Euthanasia of the American Veterinary Medical Association (AVMA).

## Cell line statement

SF9 cells for the expression of baculovirus were from Thermo Fisher (12659017, lot 421973). The tsA201 cells for the expression of AMPARs were purchased from ATCC (CRL-11268). HEK293S GnTI⁻ cells (Ric15) for electrophysiology studies were from a previously published study[59]. The cells were not authenticated experimentally for these studies and tested negative for Mycoplasma contamination.

## Reporting summary

Further information on research design is available in the Nature Portfolio Reporting Summary linked to this article.

## Data availability

The cryo-EM maps and coordinates for the NOE1–GluA1/A4-ATD and NOE1–GluA1/A4 LBD–TMD have been deposited in the Electron Microscopy Data Bank (EMDB) under accession numbers EMD-49723 and EMD-49724 and in the PDB under accession codes 9NR6 and 9NR7, respectively. The cryo-EM maps and coordinates for the GluA1/A4-ATD, GluA1/A4 LBD–TMD with 4 auxiliary proteins, and GluA1/A4 LBD–TMD with 2 TARPs have been deposited in the EMDB under accession numbers EMD-49725, EMD-49727 and EMD-49726 and in the PDB under accession codes 9NR8, 9NRA and 9NR9, respectively. The cryo-EM maps for the LBD–TMD$_{mix-4TARPs}$, LBD–TMD$_{mix-2TARPs}$, LBD–TMD$_{mix-2TARPs-2CNIHs}$, NOE1–GluA1/A4 and GluA1/A4 with ordered ATD and disordered ATD have been deposited in EMDB under accession numbers EMD-49711, EMD-49712, EMD-49713, EMD-49714, EMD-49715 and EMD-49716, respectively. The cryo-EM map for the LBD–TMD with four auxiliary subunits in the active state has been deposited in the EMDB under accession number EMD-49717. The cryo-EM maps for the GluA1/A4 LBD–TMD and noelin–GluA1/A4 LBD–TMD in the desensitized state have been deposited in the EMDB under accession numbers EMD-49718 and EMD-49719, respectively. The cryo-EM maps for the recombinant NOE1–GluA4, GluA4 with ordered ATD and disordered ATD have been deposited in the EMDB under accession numbers EMD-49722, EMD-49720 and EMD-49721, respectively. The reference models of 7LEP and 7LDD used for model building were obtained from the PDB.

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

**Acknowledgements** The authors thank R. Courtney for assistance with manuscript preparation; members of the Gouaux and Baconguis laboratories for suggestions; C. Sun for advice on cryo-EM data processing; M. Mayer and A. Matsui for their guidance and advice on electrophysiology experiments; P. Rao for help initiating the project; A. Goehring and A. Bharadwaj for maintaining the mammalian and insect cells; and A. Reddy for mass spectrometric analysis. We thank the OHSU Multiscale Microscopy Core (MMC), the cryo-EM facility at Janelia research campus, the National Center for CryoEM Access and Training (NCCAT), and the Simons Electron Microscopy Center located at the New York Structural Biology Center, supported by the NIH Common Fund Transformative High Resolution Cryo-Electron Microscopy programme (U24 GM129539, and NIGMS R24 GM154192) and by grants from the Simons Foundation (SF349247) and NY State Assembly. C.J.S. is supported by a NCI grant (CA253730). The experimental work was supported by a NIH grant to E.G. (R01NS038631). L.O.T. is supported by NIH grants NS116798 and DC004450. E.G. is an investigator of the Howard Hughes Medical Institute and thanks B. LaCroute and J. LaCroute for generous support.

**Author contributions** C.F. and E.G. designed the project. C.F. performed the sample preparation for cryo-EM and the cryo-EM data collection, data analysis and model building. C.F. performed the western blot experiments, ligand-binding assays, patch-clamp recording experiments, dNOE1 pull-down experiments and bio-layer interferometry experiments. C.F. and C.J.S. performed the SiMPull experiments and FSEC experiments. J.P. developed the initial approach for the rapid isolation of native AMPARs from rodent brain tissue. C.J.S. and N.S. prepared 7D8 and 4H9 antibodies. L.O.T. provided insights into the patch-clamp recording experiments. C.F. and E.G. wrote the initial draft of the manuscript, and all authors edited the manuscript.

**Competing interests** The authors declare no competing interests.

**Additional information**
**Correspondence and requests for materials** should be addressed to Eric Gouaux.

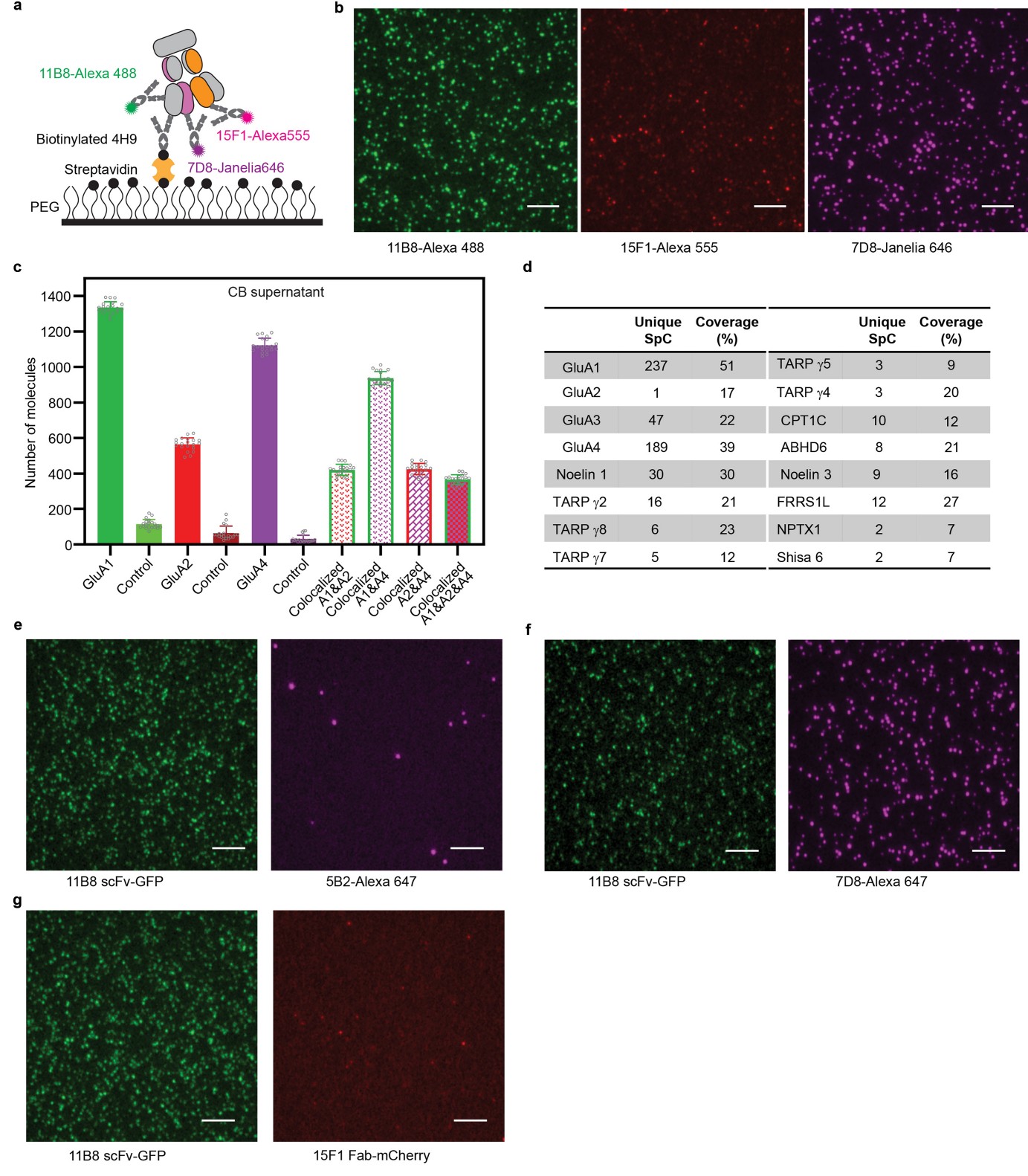

**Extended Data Fig. 1 | Cerebellar CP-AMPAR assemblies are largely composed of GluA1 and GluA4 subunits.** (a) Schematic of the SiMPull assay. 'PEG' is polyethylene glycol. (b) Fluorescence detection with the anti-GluA1 11B8-Alexa488, anti-GluA2 15F1-Alexa555, and anti-GluA4 7D8-Janelia 646 of 2% digitonin-solubilized cerebellum supernatant. The AMPARs were captured by biotinylated 4H9. Experiment was repeated independently twice with consistent results. (c) The number of GluA1, GluA2, and GluA4-containing assemblies, colocalized GluA1-GluA4 subunit-containing molecules in the cerebellum detected by SiMPull, $n = 20$ images over two independent experiments, data are shown as means ± SD. See Methods for a description of the control experiments. (d) Mass spectrometry analysis of isolated native CP-AMPAR. (e-g) SiMPull-based fluorescence detection of purified GluA1-containing CP-AMPARs with the anti-GluA1 11B8 scFv-GFP antibody, anti-GluA2 15F1 Fab-mCherry antibody, anti-GluA3 5B2-Alexa 647, and anti-GluA4 7D8-Alexa 647, captured by GFP nanobody. Experiment was repeated independently twice with consistent results, related to Fig. 1c, d.

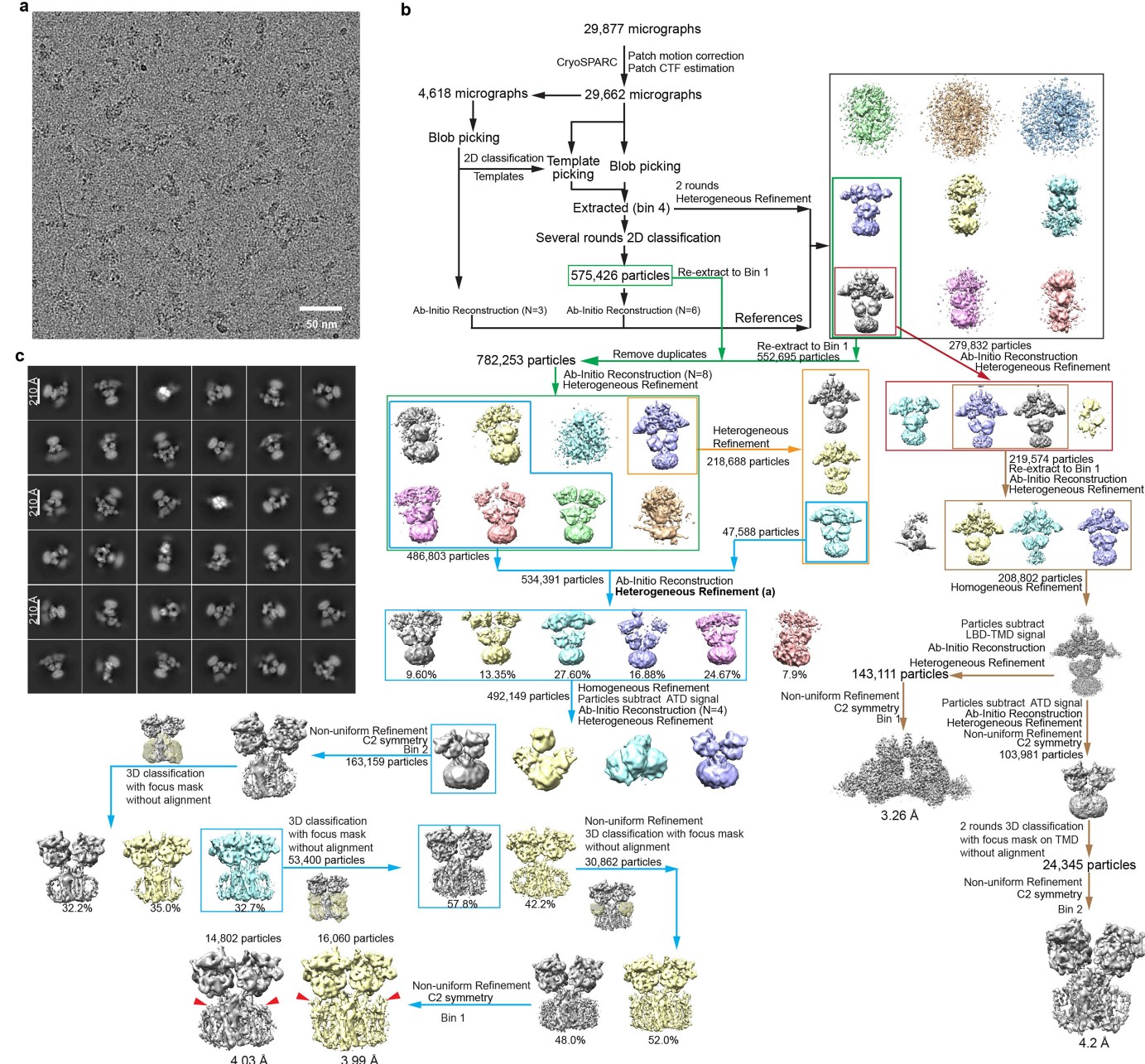

**Extended Data Fig. 2 | Cryo-EM data processing of CP-AMPARs in the closed state.** (a) Representative motion-corrected micrograph from a dataset of over 29,877 images collected in a single session. Similar micrographs were observed throughout the dataset. (b) Cryo-EM data processing workflow. The red arrow shows auxiliary subunits at the A′/C′ positions without or with extracellular protrusion. (c) Representative 2D class averages of CP-AMPARs.

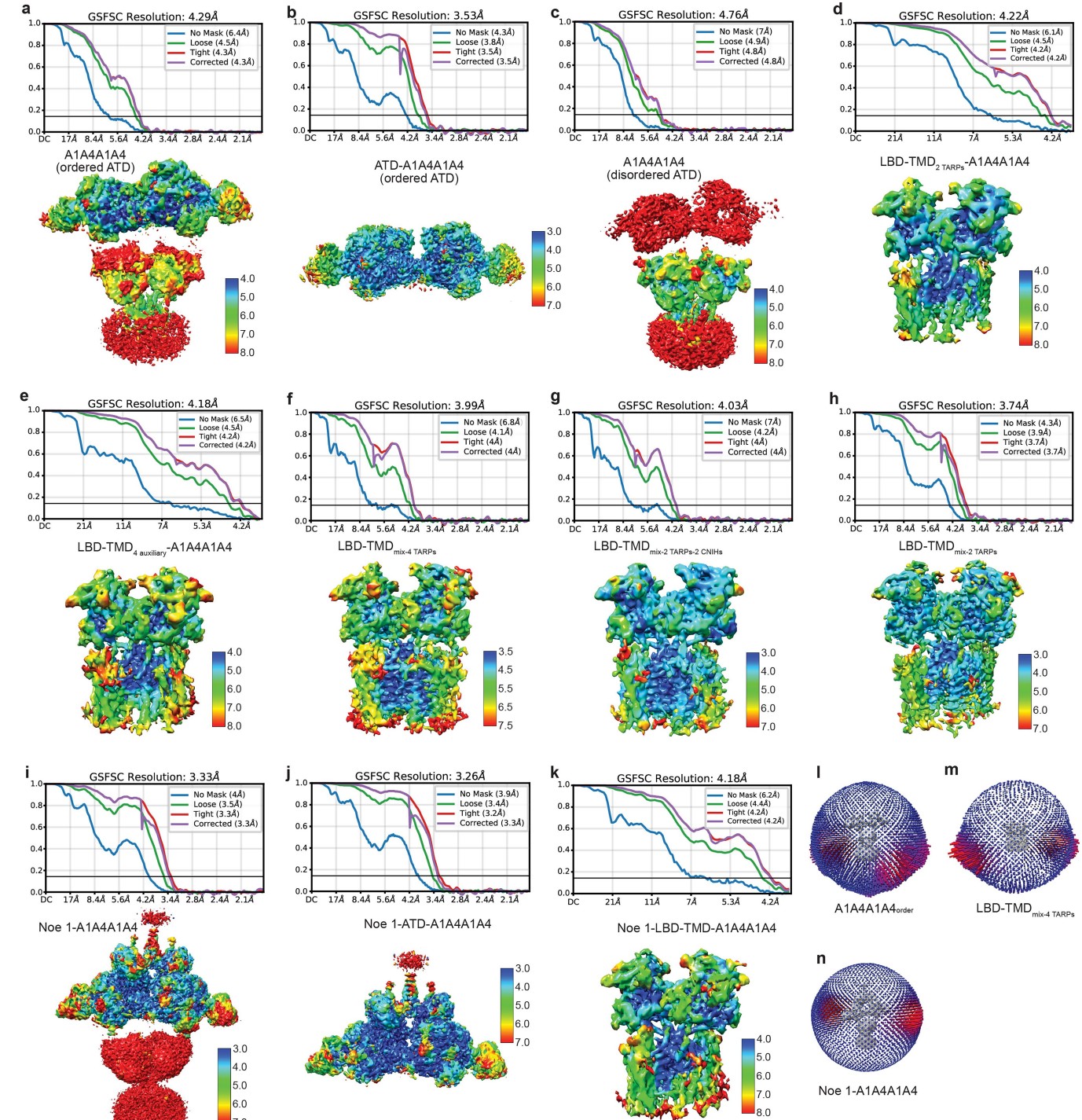

**Extended Data Fig. 3 | Cryo-EM statistics for CP-AMPARs in the closed state.** (a-k) Fourier shell correlation (FSC) curves of corresponding maps where FSC = 0.143 values (black line) were used as a cutoff. Local resolution maps are colored based on local resolution estimates as indicated by the color bar. (a) A1A4A1A4 with ordered ATD layer, (b) The ATD of A1A4A1A4, (c) A1A4A1A4 with disordered ATD layer, (d) The LBD-TMD of A1A4A1A4 with two TARPs occupying the B' and D' positions, (e) The LBD-TMD of A1A4A1A4 with four auxiliary proteins occupying the A'B'C'D' positions, (f) The LBD-TMD of all subtypes with four TARPs occupying the A'B'C'D' positions, (g) The LBD-TMD of all subtypes with two TARPs occupying the B'D' positions and assuming that two CNIHs occupy the A'C' positions, (h) The LBD-TMD of all subtypes with two TARPs occupying B'D' positions, (i) A1A4A1A4 assembly with the Noe1 complex, (j) The ATD of the A1A4A1A4-Noe1 complex, (k) the LBD-TMD of the A1A4A1A4-Noe1 complex. (l-n) Euler angle distribution of particles used for A1A4A1A4, LBD-TMD of all subtypes with four TARPs, and Noe1-A1A4A1A4 cryo-EM reconstructions.

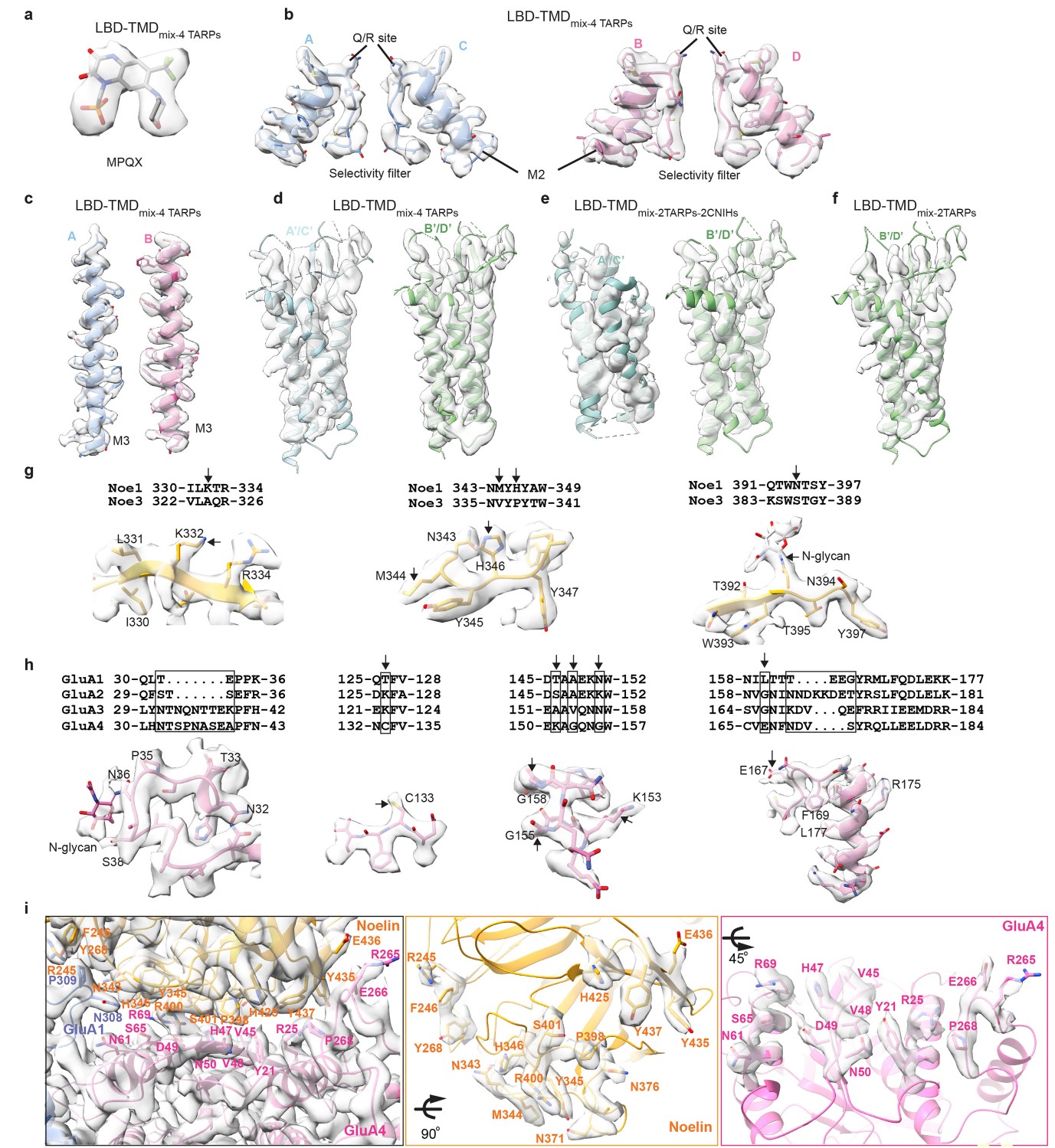

**Extended Data Fig. 4 | 'Local' density maps for reconstructions of CP-AMPARs in the closed state.** (a-d) The local maps show MPQX, the M2 helices, the selectivity filters, the M3 helices, and auxiliary protein densities in the LBD-TMD$_{mix-4TARPs}$ map, contoured at 0.09σ. (e) The local map shows auxiliary protein densities in the LBD-TMD$_{mix-2TARPs-2CNIHs}$ map, contoured at 0.044σ.

(f) The local map shows auxiliary protein densities in the LBD-TMD$_{mix-2TARPs}$ map, contoured at 0.075σ. (g-h) Cryo-EM densities of protein side chains and *N*-glycosylation were used for Noe 1 (contoured at 0.3σ) and Noe 1-bounded subunit (contoured at 0.2σ) identification. (i) Cryo-EM densities of Noe 1 and AMPAR at the interface, contoured at 0.35σ.

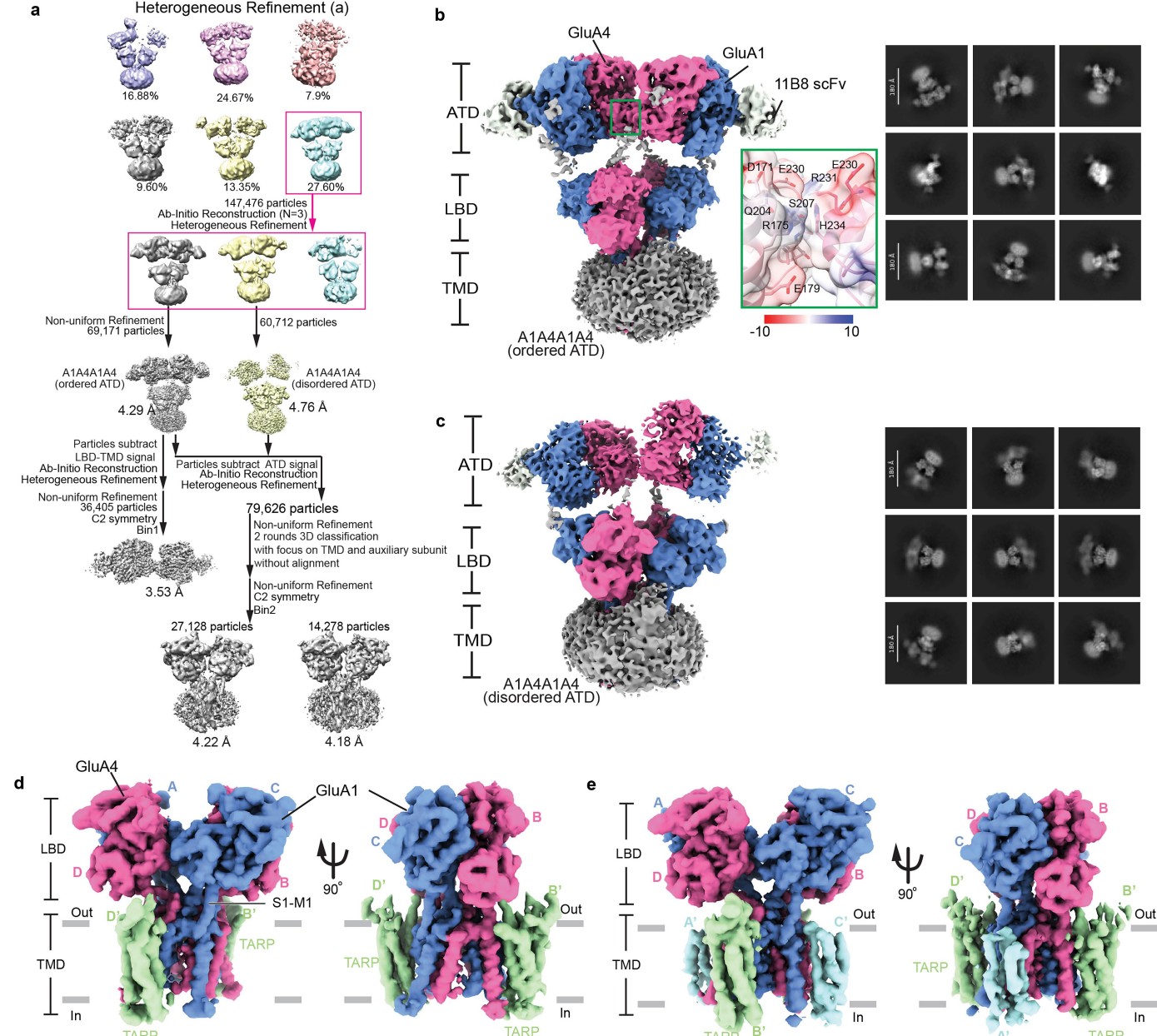

**Extended Data Fig. 5 | Cryo-EM analysis of A1A4A1A4 subtype.** (a) Cryo-EM data processing workflow for A1A4A1A4 assembly. (b-c) The cryo-EM reconstruction for the A1A4A1A4 assembly with ordered or disordered ATD layer. The insert shows electrostatic maps generated by Chimera X, depicting the dimer interface of the ordered ATD layer. The contours range from −10 kT/e (red) to +10 kT/e (blue). (d) The LBD-TMD of the A1A4A1A4 receptor with two TARPs occupying the B' and D' positions. (e) The LBD-TMD of the A1A4A1A4 receptor with four auxiliary proteins occupying the A'B'C'D' positions.

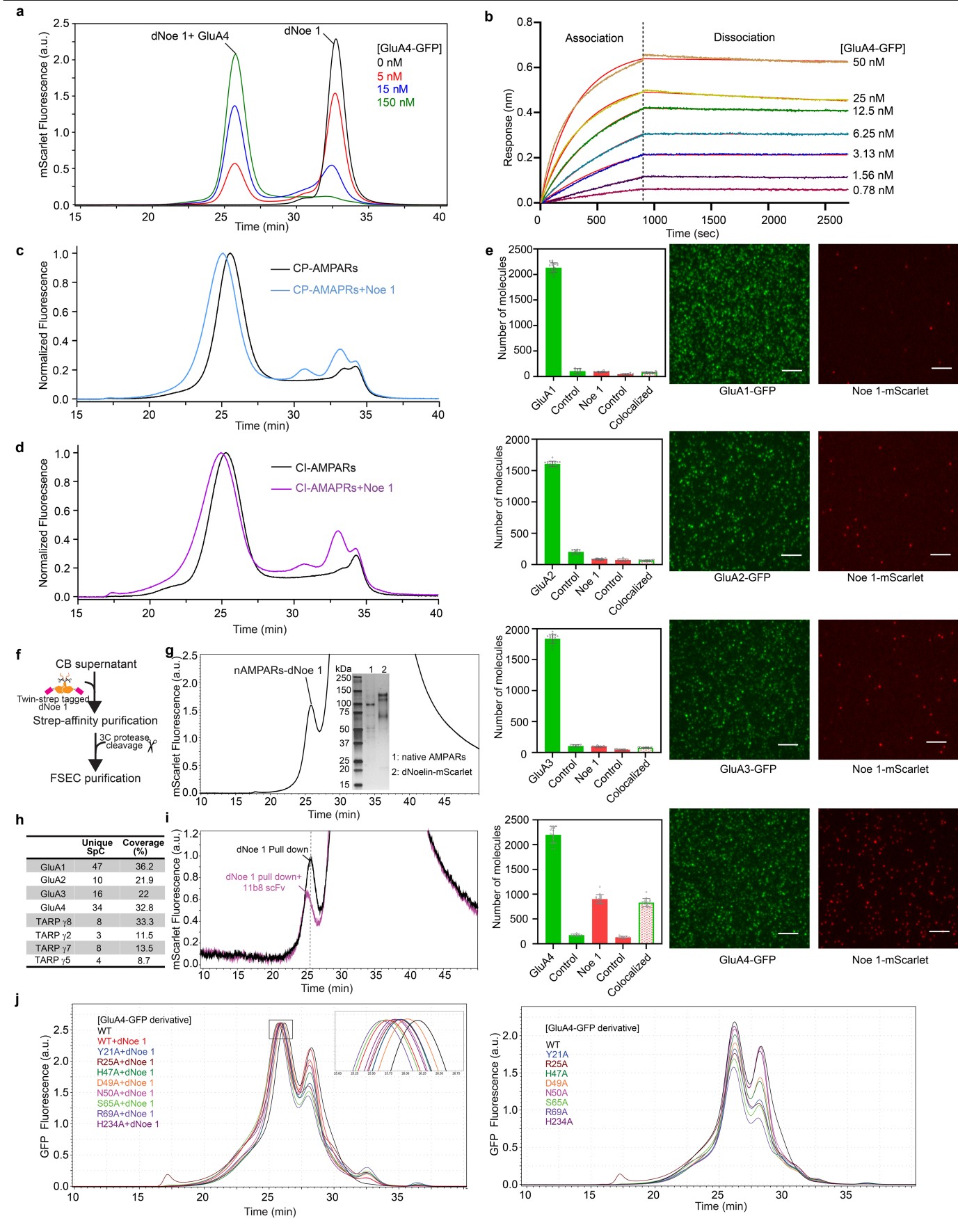

**Extended Data Fig. 6** | See next page for caption.

**Extended Data Fig. 6 | Noe1 binds preferentially with GluA4-containing receptors.** (a) The dNoe1 construct binds to recombinant homomeric GluA4 receptors as detected by FSEC. dNoe1, 5 nM. GluA4-GFP, 0–150 nM. (b) Octet measurements of dNoe1 binding to homomeric GluA4, the Kd is ~9.4 nM. Concentrations of the rGluA4-GFP ranging from 0.78 nM to 50 nM were applied. The fitting curves are colored red. (c) 500 nM full-length Noe1 incubated with 10 nM CP-AMPARs, visualized by FSEC. (d) 500 nM full-length Noe1 incubated with native GluA2-containing AMPARs, visualized by FSEC. (e) Full-length Noe1 prefers to bind with GluA4 detected by SiMPull, n = 20, 19, 18, and 15 images for GluA1, A2, A3 and A4, respectively, and shown as means ± SD, examined over two independent experiments. (f-i) dNoe1 pull-down AMPARs from rat cerebellum, (f) Flowchart for dNoe1 pull-down assay, (g) The FSEC profile of dNoe1 pull-down complexes, from single experiment. (h) Mass spectrometry analysis of dNoe1 pull-down complexes, (i) 11B8 scFv can shift dNoe1 pull-down complexes. (j) FSEC profiles of 10 nM GluA4 mutants with 50 nM dNoe 1-mScarlet or alone, relative to Fig. 3d. For each GluA4 mutant, we measured the peak shift after incubation with dNoe1 based on these FSEC traces. This is the experimental data that is then summarized in the main text Fig. 3d.

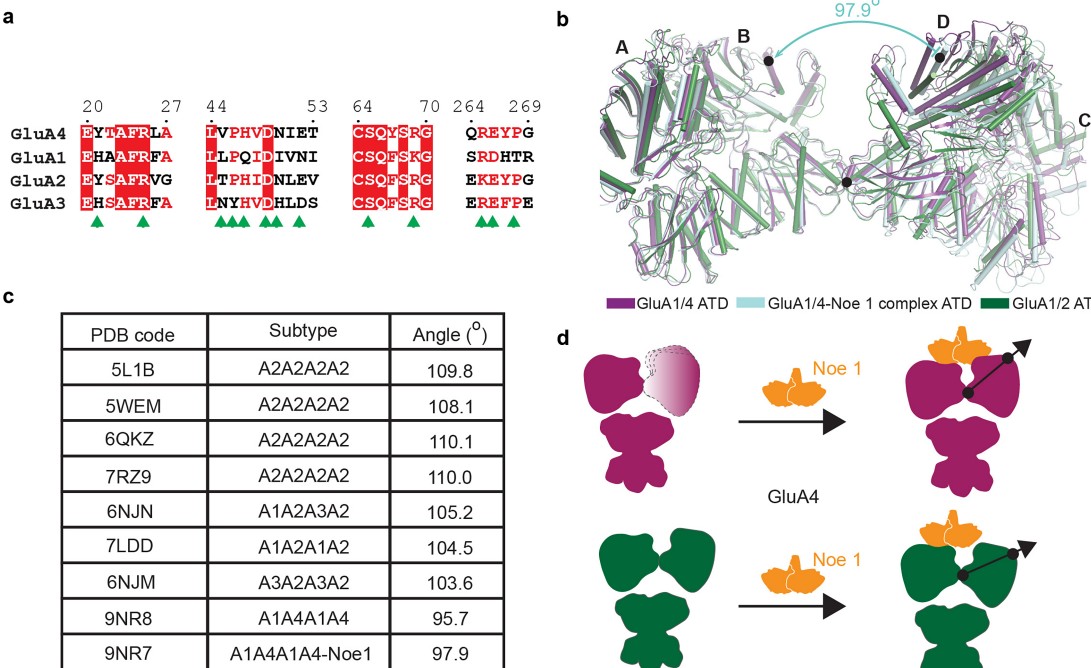

**a**

```
            20      27   44        53    64    70 264 269
GluA4   EYTAFRLA   IVPHVDNIET   CSQYSRG   QREYPG
GluA1   EHAAFRFA   LLPQIDIVNI   CSQESKG   SRDHTR
GluA2   EYSAFRVG   LTPHIDNLEV   CSQFSRG   EKEYPG
GluA3   EHSAFRFA   LNYHVDHLDS   CSQFSRG   EREFPE
```

**c**

| PDB code | Subtype | Angle (°) |
|----------|---------|-----------|
| 5L1B | A2A2A2A2 | 109.8 |
| 5WEM | A2A2A2A2 | 108.1 |
| 6QKZ | A2A2A2A2 | 110.1 |
| 7RZ9 | A2A2A2A2 | 110.0 |
| 6NJN | A1A2A3A2 | 105.2 |
| 7LDD | A1A2A1A2 | 104.5 |
| 6NJM | A3A2A3A2 | 103.6 |
| 9NR8 | A1A4A1A4 | 95.7 |
| 9NR7 | A1A4A1A4-Noe1 | 97.9 |

**Extended Data Fig. 7 | Differences in amino acid sequences and domain positions modulate Noe1 binding.** (a) The sequence alignment of GluA1-4 subunits for residues at the Noe1 binding surface. (b) Superimposing structures of the GluA1/A4 ATD, GluA1/A4-Noe1 complex ATD, and GluA1/A2 ATD, aligned at the B position. (c) The angle of the ATD dimers of different GluA2-containing AMPAR subtypes, the center of mass of helices A at B or D positions, respectively, and helices G at B and D positions interface for the angle determined was labeled in the black dot. (d) The cartoon shows the differences between GluA1/A4 and GluA1/A2 when binding with Noe1. Noe1 interacts with GluA2 receptors, only one protomer can bind to an ATD dimer, resulting in a potentially unstable interaction. For further exploration of Noe1 interactions with GluA1-4 receptors, see Supplementary Fig. 4.

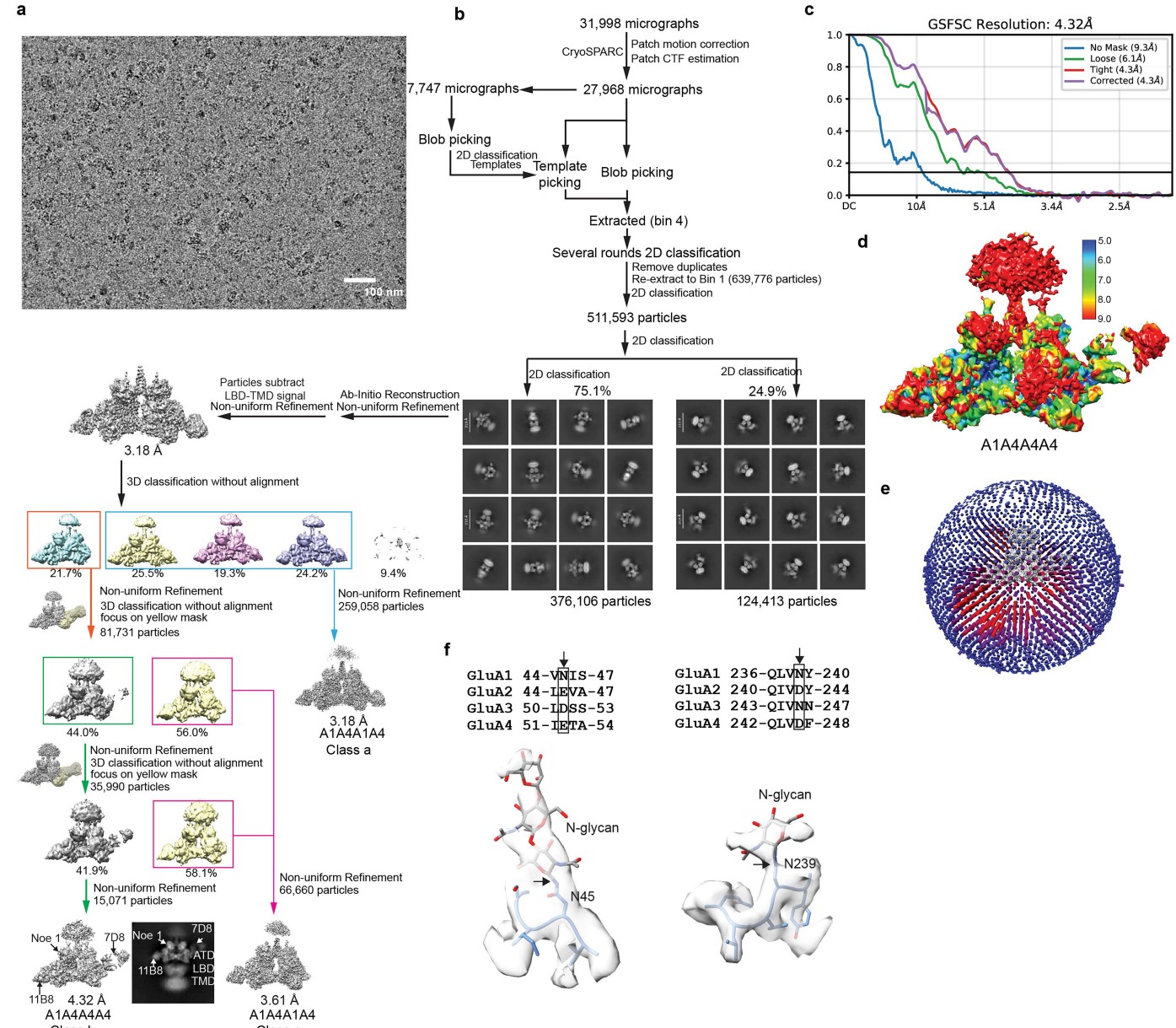

**Extended Data Fig. 8 | Cryo-EM data processing for CP-AMPARs with recombinant Noe 1.** (a) Representative motion-corrected micrograph from a dataset of over 31,998 images collected in a single session. Similar micrographs were observed throughout the dataset. (b) Cryo-EM data processing workflow. (c) The FSC curves of the A1A4A4A4 subtype (Class b). (d) The local resolution estimation of A1A4A4A4. (e) Euler angle distribution of particles of the A1A4A4A4 subtype. (f) Cryo-EM densities of protein side chains and *N*-glycosylation were used for Class c subunit identification in the absence of additional information, such as from antibody fragment binding or from Noe 1. The unique *N*-glycosylation site is consistent with this subunit being GluA1.

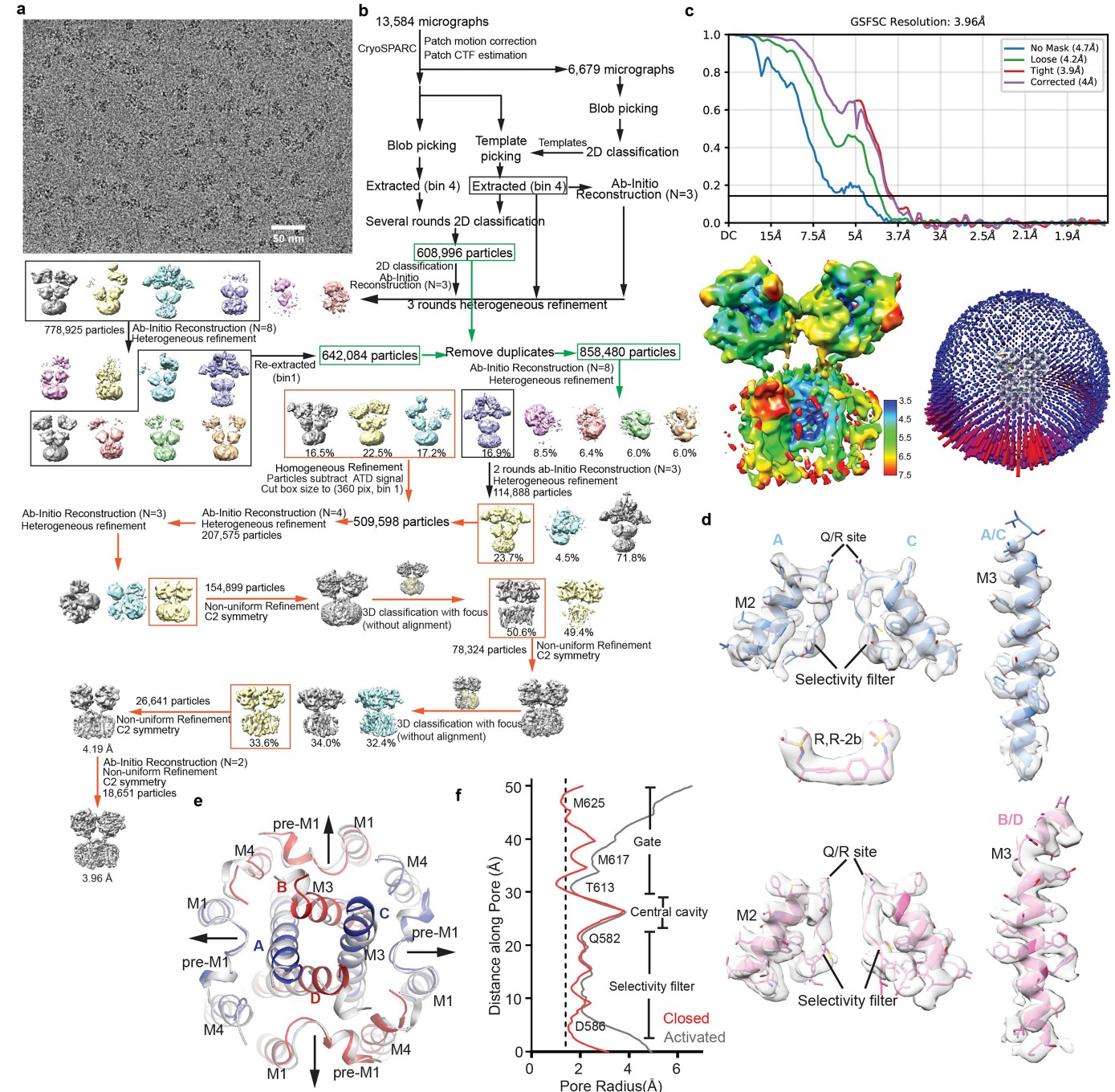

**Extended Data Fig. 9 | Cryo-EM data processing of CP-AMPARs in active state.** (a) Representative motion-corrected micrograph from a dataset of over 13,584 images collected in a single session. Similar micrographs were observed throughout the dataset. (b) Cryo-EM data processing workflow. (c) The FSC curves, local resolution estimation, and Euler angle distribution of the LBD-TMD from all subtypes. (d) The local density map for M2 helices, selectivity filter, and M3 helices in active state. (e) The top view of superposed CP-AMPARs with four auxiliary proteins in the closed (colored) and active states (grey). (f) Pore radius in the closed and activated state of CP-AMPARs. Dashed line represents the 1.4 Å radius of a water molecule.

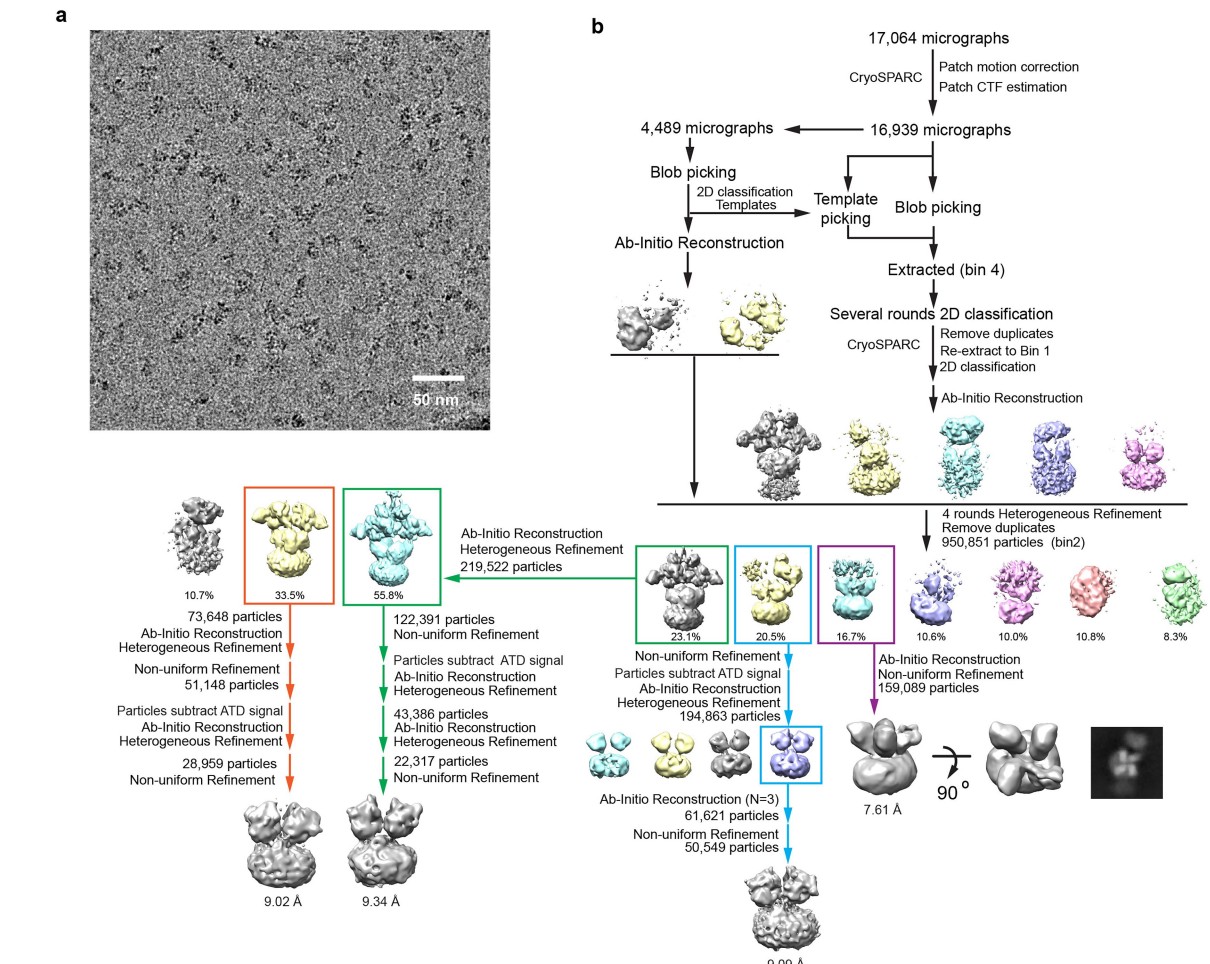

**Extended Data Fig. 10 | Cryo-EM data processing of CP-AMPARs in desensitized states.** (a) Representative motion-corrected micrograph from a dataset of over 17,064 images collected in a single session. Similar micrographs were observed throughout the dataset. (b) Cryo-EM data processing workflow.

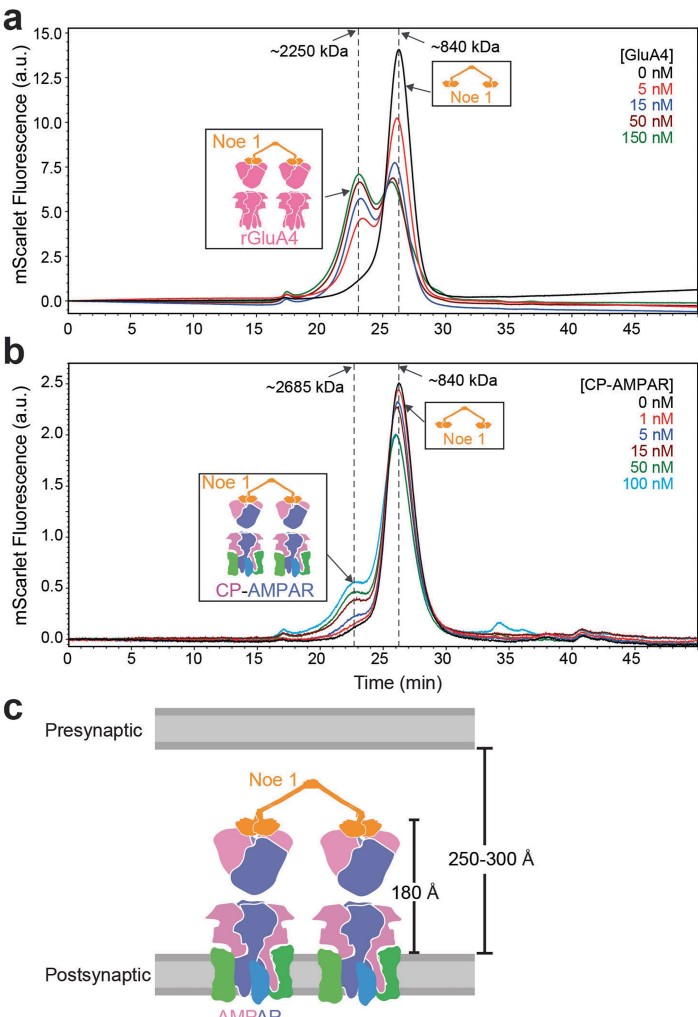

**Extended Data Fig. 11 | Noe 1 clusters AMPARs via dimerization.** (a) FSEC results for Noe 1-mScarlet (5 nM) with recombinant GluA4, detecting mScarlet fluorescence. (b) FSEC results for Noe 1-mScarlet (5 nM) with cerebellar CP-AMPARs, detecting mScarlet fluorescence. The molecular weights of Noe 1 (~840 kDa), GluA4 (~568 kDa), and CP-AMPARs (~1180 kDa) were calibrated using a standard marker (Supplementary Fig. 8). (c) The cartoon shows how tetrameric Noe 1 can bridge two AMPARs and promote dimerization within the context of the synaptic cleft, leaving ample room for the binding of additional synaptic proteins to Noe 1.

# Extended Data Table 1 | Cryo-EM data collection, refinement and validation statistics

| | All subtypes combined LBD-TMD$_{mix}$ with 4 TARPs (EMDB-49711) | All subtypes combined LBD-TMD$_{mix}$ with 2 TARPs (EMDB-49712) | All subtypes combined LBD-TMD$_{mix}$ with 2 TARPs 2CNIH (EMDB-49713) | Noe 1-GluA1/4 (EMDB-49714) | Noe 1-GluA1/4-ATD (EMDB-49723) (PDB 9NR6) | Noe 1-GluA1/4 LBD-TMD (EMDB-49724) (PDB 9NR7) | GluA1/4 with ordered ATD (EMDB-49715) | GluA1/4 with disordered ATD (EMDB-49716) | GluA1/4 ATD (EMDB-49725) (PDB 9NR8) | GluA1/4 LBD-TMD with 4 auxiliary proteins (EMDB-49727) (PDB 9NRA) | GluA1/4 LBD-TMD with 2 TARPs (EMDB-49726) (PDB 9NR9) |
|---|---|---|---|---|---|---|---|---|---|---|---|
| **Data collection and processing** | | | | | | | | | | | |
| Magnification | | | | | | 130,000x | | | | | |
| Voltage (kV) | | | | | | 300 | | | | | |
| Electron exposure (e–/Å$^2$) | | | | | | 50 | | | | | |
| Defocus range (µm) | | | | | | -1.2 to -2.2 | | | | | |
| Pixel size (Å) | | | | | | 0.94 | | | | | |
| Symmetry imposed | C2 | C2 | C2 | C1 | C2 | C2 | C1 | C1 | C2 | C2 | C2 |
| Initial particle images (no.) | 700,951 | 700,951 | 700,951 | 700,951 | 700,951 | 700,951 | 700,951 | 700,951 | 700,951 | 700,951 | 700,951 |
| Final particle images (no.) | 16,060 | 56,662 | 14,802 | 208,802 | 143,111 | 24,345 | 69,171 | 60,712 | 36,405 | 14,278 | 27,128 |
| Map resolution (Å) | 3.99 | 3.74 | 4.03 | 3.33 | 3.26 | 4.18 | 4.29 | 4.76 | 3.53 | 4.18 | 4.22 |
| FSC threshold | 0.143 | 0.143 | 0.143 | 0.143 | 0.143 | 0.143 | 0.143 | 0.143 | 0.143 | 0.143 | 0.143 |
| Map resolution range (Å) | 2.0-8.4 | 3.1-8.8 | 2.4- 8.2 | 2.9-54.3 | 2.0-7.6 | 4.2-10.6 | 2.0-9.9 | 4.0-10.5 | 2.0-8.3 | 4.2-10.6 | 4.2- 10.0 |
| Map sharpening B factor (Å$^2$) | 74.4 | 96.7 | 62.0 | 85.7 | 89.3 | 88.5 | 91.4 | 159.4 | 90.7 | 87.9 | 117.0 |
| **Refinement** | | | | | | | | | | | |
| Initial model used (PDB code) | | | | | 7LDD | 7LEP | | | 7LDD | 7LEP | 7LEP |
| Model resolution (Å) | | | | | 3.4 | 4.7 | | | 3.8 | 4.6 | 4.5 |
| FSC threshold | | | | | 0.5 | 0.5 | | | 0.5 | 0.5 | 0.5 |
| Model composition | | | | | | | | | | | |
| Non-hydrogen atoms | | | | | 19239 | 15603 | | | 14602 | 15639 | 13720 |
| Protein residues | | | | | 2512 | 2190 | | | 1958 | 2195 | 1865 |
| Ligands | | | | | 34 | 4 | | | 22 | 4 | 4 |
| B factors (Å$^2$) | | | | | | | | | | | |
| Protein | | | | | 41.79 | 115.87 | | | 34.06 | 131.86 | 134.20 |
| Ligand | | | | | 69.77 | 69.26 | | | 52.17 | 93.69 | 97.77 |
| R.m.s. deviations | | | | | | | | | | | |
| Bond lengths (Å) | | | | | 0.008 | 0.003 | | | 0.009 | 0.003 | 0.003 |
| Bond angles (°) | | | | | 0.646 | 0.784 | | | 0.834 | 0.834 | 0.841 |
| Validation | | | | | | | | | | | |
| MolProbity score | | | | | 1.41 | 1.68 | | | 1.58 | 1.78 | 1.70 |
| Clashscore | | | | | 4.39 | 5.81 | | | 4.33 | 7.44 | 6.22 |
| Poor rotamers(%) | | | | | 0.11 | 0.07 | | | 0.31 | 0.00 | 0.00 |
| Ramachandran plot | | | | | | | | | | | |
| Favored (%) | | | | | 96.83 | 94.73 | | | 94.69 | 94.51 | 94.87 |
| Allowed (%) | | | | | 3.17 | 5.27 | | | 5.31 | 5.49 | 5.13 |
| Disallowed (%) | | | | | 0.00 | 0.00 | | | 0.00 | 0.00 | 0.00 |

| | All subtypes combined LBD-TMD with 4 auxiliary proteins in active state (EMDB-49717) | GluA1/4 LBD-TMD in desensitized state (EMDB-49718) | Noe 1-GluA1/4 LBD-TMD in desensitized state (EMDB-49719) | rGluA4 with ordered ATD (EMDB-49720) | rGluA4 with disordered ATD (EMDB-49721) | Noe 1-rGluA4 (EMDB-49722) |
|---|---|---|---|---|---|---|
| **Data collection and processing** | | | | | | |
| Magnification | 105,000x | 130,000x | 130,000x | 105,000x | 105,000x | 105,000x |
| Voltage (kV) | 300 | 300 | 300 | 300 | 300 | 300 |
| Electron exposure (e–/Å$^2$) | 50 | 50 | 50 | 60 | 60 | 60 |
| Defocus range (µm) | -1.2 to -2.2 | -1.2 to -2.2 | -1.2 to -2.2 | -1.8 to -2.5 | -1.8 to -2.5 | -1.8 to -2.5 |
| Pixel size (Å) | 0.831 | 0.94 | 0.94 | 1.196 | 1.196 | 1.196 |
| Symmetry imposed | C2 | C1 | C1 | C1 | C1 | C1 |
| Initial particle images (no.) | 858,480 | 950,851 | 950,851 | 251,988 | 251,988 | 251,988 |
| Final particle images (no.) | 26,641 | 28,959 | 22,317 | 44,360 | 54,376 | 34,395 |
| Map resolution (Å) | 3.96 | 9.02 | 9.34 | 8.02 | 4.61 | 7.44 |
| FSC threshold | 0.143 | 0.143 | 0.143 | 0.143 | 0.143 | 0.143 |
| Map resolution range (Å) | 2.0-9.7 | 9.0-17.0 | 9.0-17.0 | 7.0-16.9 | 3.9-11.0 | 6.6-20.0 |

Summary of microscope settings, data processing parameters, map resolution, and model refinement and validation metrics for each dataset and corresponding structure.

# Reporting Summary

## Statistics

For all statistical analyses, confirm that the following items are present in the figure legend, table legend, main text, or Methods section.

| n/a | Confirmed | |
|---|---|---|
| ☐ | ☒ | The exact sample size (*n*) for each experimental group/condition, given as a discrete number and unit of measurement |
| ☐ | ☒ | A statement on whether measurements were taken from distinct samples or whether the same sample was measured repeatedly |
| ☐ | ☒ | The statistical test(s) used AND whether they are one- or two-sided<br>*Only common tests should be described solely by name; describe more complex techniques in the Methods section.* |
| ☒ | ☐ | A description of all covariates tested |
| ☐ | ☒ | A description of any assumptions or corrections, such as tests of normality and adjustment for multiple comparisons |
| ☐ | ☒ | A full description of the statistical parameters including central tendency (e.g. means) or other basic estimates (e.g. regression coefficient) AND variation (e.g. standard deviation) or associated estimates of uncertainty (e.g. confidence intervals) |
| ☐ | ☒ | For null hypothesis testing, the test statistic (e.g. *F*, *t*, *r*) with confidence intervals, effect sizes, degrees of freedom and *P* value noted<br>*Give P values as exact values whenever suitable.* |
| ☒ | ☐ | For Bayesian analysis, information on the choice of priors and Markov chain Monte Carlo settings |
| ☒ | ☐ | For hierarchical and complex designs, identification of the appropriate level for tests and full reporting of outcomes |
| ☒ | ☐ | Estimates of effect sizes (e.g. Cohen's *d*, Pearson's *r*), indicating how they were calculated |

*Our web collection on statistics for biologists contains articles on many of the points above.*

## Software and code

Policy information about availability of computer code

| Data collection | SerialEM 3.7, pClamp 10, Octet software 10.0 (Sartorius) |
|---|---|
| Data analysis | GraphPad Prism 10, CryoSPARC v4.4, COOT 0.9, Phenix 1.20, Chimera 1.16, Pymol 3.1, ChimeraX 1.6, Clamp fit 11.2, Octet software 10.0 (Sartorius) |

For manuscripts utilizing custom algorithms or software that are central to the research but not yet described in published literature, software must be made available to editors and reviewers. We strongly encourage code deposition in a community repository (e.g. GitHub). See the Nature Portfolio guidelines for submitting code & software for further information.

## Data

Policy information about availability of data

All manuscripts must include a data availability statement. This statement should provide the following information, where applicable:
- Accession codes, unique identifiers, or web links for publicly available datasets
- A description of any restrictions on data availability
- For clinical datasets or third party data, please ensure that the statement adheres to our policy

The cryo-EM maps and coordinates for the Noelin-GluA1/A4-ATD and Noelin-GluA1/A4 LBD-TMD have been deposited in the Electron Microscopy Data Bank (EMDB) under accession numbers EMD-49723 and EMD-49724 and in the Protein Data Bank (PDB) under accession codes 9NR6 and 9NR7, respectively. The cryo-EM maps and coordinates for the GluA1/A4-ATD, GluA1/A4 LBD-TMD with 4 auxiliary proteins, and GluA1/A4 LBD-TMD with 2 TARPs have been deposited in EMDB

under accession numbers EMD-49725, EMD-49727, and EMD-49726 and in PDB under accession codes 9NR8, 9NRA, and 9NR9, respectively. The cryo-EM maps for the LBD-TMDmix-4TARPs, LBD-TMDmix-2TARPs, LBD-TMDmix-2TA.RPs-2CNIHs, Noelin-GluA1/A4, GluA1/A4 with ordered ATD and disordered ATD have been deposited in EMDB under accession numbers EMD-49711, EMD-49712, EMD-49713, EMD-49714, EMD-49715, and EMD-49716, respectively. The cryo-EM map for the LBD-TMD with 4 auxiliary subunits in the active state has been deposited in EMDB under accession numbers EMD-49717. The cryo-EM maps for the GluA1/A4 LBD-TMD and Noelin-GluA1/A4 LBD-TMD in the desensitized state have been deposited in the EMDB under accession numbers EMD-49718 and EMD-49719, respectively. The cryo-EM maps for the recombinant Noelin-GluA4, GluA4 with ordered ATD and disordered ATD have been deposited in EMDB under accession numbers EMD-49722, EMD-49720, and EMD-49721, respectively. The reference models of 7LEP and 7LDD used for model building were obtained from the PDB.

# Research involving human participants, their data, or biological material

Policy information about studies with human participants or human data. See also policy information about sex, gender (identity/presentation), and sexual orientation and race, ethnicity and racism.

| | |
|---|---|
| Reporting on sex and gender | N/A |
| Reporting on race, ethnicity, or other socially relevant groupings | N/A |
| Population characteristics | N/A |
| Recruitment | N/A |
| Ethics oversight | N/A |

Note that full information on the approval of the study protocol must also be provided in the manuscript.

# Field-specific reporting

Please select the one below that is the best fit for your research. If you are not sure, read the appropriate sections before making your selection.

☒ Life sciences ☐ Behavioural & social sciences ☐ Ecological, evolutionary & environmental sciences

For a reference copy of the document with all sections, see nature.com/documents/nr-reporting-summary-flat.pdf

# Life sciences study design

All studies must disclose on these points even when the disclosure is negative.

| | |
|---|---|
| Sample size | For cryo-EM experiments, sample sizes were determined by the availability of microscope time and the number of high-quality particles obtained. The final particle numbers were sufficient to achieve the reported resolution, as validated by Fourier shell correlation (FSC) analysis. For single-molecule pull-down (SiMPull) experiments, at least 15 images were collected, which provided reproducible results across replicates. Electrophysiology recordings were repeated at least 14 times using different cells. The sample sizes for both SiMPull and electrophysiology experiments were determined based on the observed consistency and variability of the data, informed by prior literature and our previous experience. Overall, the sample sizes were sufficient to support the robustness and reproducibility of the reported findings. |
| Data exclusions | No data were excluded from the analyses. |
| Replication | Cryo-EM–related experiments, including protein purification and FSEC, were independently reproduced at least three times with consistent results. SDS-PAGE, Western blot, and mass spectrometry analyses were independently repeated twice, yielding reproducible outcomes. Electrophysiology recordings were performed on at least 14 different cells, and all replicates produced consistent responses. Radioligand binding assays were carried out in three parallel trials, all of which yielded comparable binding profiles. Octet experiments were independently repeated three times, with consistent kinetic parameters observed across replicates. SiMPull experiments were performed by collecting at least 17 images, and all replicates yielded reproducible patterns. |
| Randomization | For cryo-EM data processing, particle datasets were randomly divided into two halves following the standard gold-standard refinement procedure implemented in cryoSPARC. Group allocation was not applicable to SiMPull experiments, cryo-EM–related biochemical experiments, electrophysiology, Octet experiments, and radioligand binding assays, as these experiments did not involve predefined experimental groups or treatment conditions. For SiMPull experiments, images were acquired from randomly selected regions of the sample chamber to avoid selection bias. Cryo-EM–related biochemical experiments, including protein purification and FSEC, were repeated using independently prepared samples from different batches of rat cerebellum. For electrophysiology recordings, GFP-positive cells were randomly selected for patching, without prior knowledge of their electrophysiological properties. Octet and radioligand binding assays were performed on uniform sample preparations and thus did not require further group allocation. |
| Blinding | The investigators were not blinded. Blinding was not applicable to cryo-EM data collection and analysis, cryo-EM related biochemical experiments, and mass spectrometry analyses, because this type of study does not use group allocation. It is not technically or practically feasible to do so for electrophysiology recording, Octet experiments, FSEC experiments, radioligand binding assays or Single molecule pull down experiments. |

# Reporting for specific materials, systems and methods

We require information from authors about some types of materials, experimental systems and methods used in many studies. Here, indicate whether each material, system or method listed is relevant to your study. If you are not sure if a list item applies to your research, read the appropriate section before selecting a response.

## Materials & experimental systems

| n/a | Involved in the study |
|-----|----------------------|
| ☐ | ☒ Antibodies |
| ☐ | ☒ Eukaryotic cell lines |
| ☒ | ☐ Palaeontology and archaeology |
| ☐ | ☒ Animals and other organisms |
| ☒ | ☐ Clinical data |
| ☒ | ☐ Dual use research of concern |
| ☒ | ☐ Plants |

## Methods

| n/a | Involved in the study |
|-----|----------------------|
| ☒ | ☐ ChIP-seq |
| ☒ | ☐ Flow cytometry |
| ☒ | ☐ MRI-based neuroimaging |

## Antibodies

| Antibodies used | 11B8 scFv anti-GluA1 (produced by our lab); 15F1 Fab anti-GluA2 (produced by our lab); 5B2 Fab anti-GluA3 (produced by our lab);7D8 anti-GluA4 (produced by our lab), 4H9 anti-GluA1 (produced by our lab), L21-32R Fab anti-GluA2 (DNA sequence from addgene, https://www.addgene.org/177480/), anti-GFP nanobody (plasmid of the GFP nanobody was a gift from Brett Collins, and was expressed, purified in our lab). Commercial antibodies: anti-GluA1 (Millipore, N453, 04-823,), anti-GluA2 ( Thermo Fisher, N/A, PAS-19496), anti-GluA3 (Invitrogen,3B3,32-0400), anti-GluA4 ( Millipore, N/A, ab1508). We have stated in the method that the antibodies used for structural determination were not diluted, antibodies used for single molecule pull down experiments were diluted to at a concentration of 10 to 30 ug/ml and antibodies used for Western Blot were diluted in1: 1000. IRDye 800 CW anti-mouse/rabbit secondary antibodies were used for western blot visualization. Blots were developed by adding secondary antibodies at a ratio of 1:10,000. |
|-----------------|-----------------------------------------------------------------------------------------------------------------------------------|
| Validation | Validation of 11B8 scFv anti-GluA1, 5F1 Fab anti-GluA2 and 5B2 Fab anti-GluA3 used for cryo-EM structure determination and single molecule pull down experiments can be found in the previous published literature (Zhao, Y. et al. Architecture and subunit arrangement of native AMPA receptors elucidated by cryo-EM. Science 364, 355-362 (2019)). The validation of anti-GluA1 4H9 and anti-GluA4 7D8 could be found in the Supplementary Fig. 1. The validation of L21-32R anti-GluA2 could be found in the website:https://www.addgene.org/177480/. The refernece for the GFP nanobody is : Kubala, M. et al. Structural and thermodynamic analysis of the GFP:GFP-nanobody complex. Protein Sci. 2010 Dec;19(12):2389-401. The validation of commercial antibodies anti-GluA1, anti-GluA2, anti-GluA3, and anti-GluA4 for western blot could be found in the websites: https://www.emdmillipore.com/US/en/product/Anti-phospho-GluR1-Ser831-Antibody-clone-N453-rabbit-monoclonal, MM_NF-04-823?ReferrerURL=https%3A%2F%2Fwww.google.com%2F; https://www.thermofisher.com/antibody/product/GluR2-Antibody-Polyclonal/PAS-19496; https://www.thermofisher.com/antibody/product/GluR3-Antibody-clone-3B3-Monoclonal/32-0400; https://www.emdmillipore.com/US/en/product/Anti-Glutamate-Receptor-4-Antibody, MM_NF-AB1508, respectively. |

## Eukaryotic cell lines

Policy information about cell lines and Sex and Gender in Research

| Cell line source(s) | Sf9 cells for expression of Baculovirus are from Thermofisher (12659017, lot 421973). HEK2935S GnTI- cells for protein expression and electrophysiology studies were purchased from ATCC (CRL-3022). The tsA201 cells for the expression of AMPARs are purchase form ATCC (CRL-11268). |
|---------------------|--------------------------------------------------------------------------------------------------------------------------------------------------------------------|
| Authentication | The cells were routinely maintained in our lab. They were not authenticated experimentally for these studies. |
| Mycoplasma contamination | Sf9 cells,tsA 201 cells and HEK293S GnTI- cells were tested negative. |
| Commonly misidentified lines (See ICLAC register) | No commonly misidentified lines were applied. |

## Animals and other research organisms

Policy information about studies involving animals; ARRIVE guidelines recommended for reporting animal research, and Sex and Gender in Research

| Laboratory animals | The study did not involve laboratory animals, rat carcasses donated from other laboratories of OHSU. |
|--------------------|------------------------------------------------------------------------------------------------------|
| Wild animals | The study did not involve wild animals. |
| Reporting on sex | The cerebellum tissue derived from male and female Rattus norvegicus rats |

| Field-collected samples | No filed-collected samples were used in this study. |
|---|---|
| Ethics oversight | All rats were euthanized under the OHSU Institutional Animal Care and Use Committee (IACUC) protocols, consistent with the recommendations of the Panel on Euthanasia of the American Veterinary Medical Association (AVMA). |

Note that full information on the approval of the study protocol must also be provided in the manuscript.

## Plants

| Seed stocks | *Report on the source of all seed stocks or other plant material used. If applicable, state the seed stock centre and catalogue number. If plant specimens were collected from the field, describe the collection location, date and sampling procedures.* |
|---|---|
| Novel plant genotypes | *Describe the methods by which all novel plant genotypes were produced. This includes those generated by transgenic approaches, gene editing, chemical/radiation-based mutagenesis and hybridization. For transgenic lines, describe the transformation method, the number of independent lines analyzed and the generation upon which experiments were performed. For gene-edited lines, describe the editor used, the endogenous sequence targeted for editing, the targeting guide RNA sequence (if applicable) and how the editor was applied.* |
| Authentication | *Describe any authentication procedures for each seed stock used or novel genotype generated. Describe any experiments used to assess the effect of a mutation and, where applicable, how potential secondary effects (e.g. second site T-DNA insertions, mosiacism, off-target gene editing) were examined.* |

