## [Peer Review File · Nature]

Gating and Noelin-clustering of native Ca^{2+} -permeable AMPA receptors

Corresponding Author: Dr Eric Gouaux

Version 0:

Reviewer comments:

Referee #1

(Remarks to the Author)

AMPA receptors are the primary mediators of fast excitatory synaptic transmission in the mammalian brain and are essential for synaptic plasticity, learning, and memory. Most AMPARs incorporate the GluA2 subunit, which undergoes RNA editing at the Q/R site to render the receptor impermeable to Ca^{2+} . In contrast, Ca^{2+} -permeable AMPARs (CP-AMPA receptors) lack the edited GluA2 subunit and are highly permeable to calcium. CP-AMPA receptors have unique biophysical and pharmacological properties and play critical roles in various forms of synaptic plasticity, including long-term potentiation, long-term depression, and homeostatic plasticity. Their dysregulation has been implicated in numerous neurological disorders, including epilepsy and stroke. Despite their importance, the biophysical studies of CP-AMPA receptors have been limited. There is no knowledge about the native architecture, subunit composition, and auxiliary protein associations of CP-AMPA receptors. This manuscript by Fang et al. represents a major advance in this area. The authors purified native CP-AMPA receptors from rat cerebellum, determined cryo-EM structures, revealing their subunit composition, auxiliary proteins, and association with the synaptic regulator Noelin 1. The authors identified a unique interaction site of Noelin 1 at the GluA4 B/D positions in a novel extracellular pose, stabilizing the ATD layer without altering receptor gating. The binding site of Noelin 1 on AMPAR has never been reported before although Noelin has been reported in past studies to associate with AMPA receptors and modulate synaptic properties. This configuration of Noelin 1 on AMPAR complex looks like a “unicorn horn”, stabilizing the ATD layer, adding new insights into how extracellular regulators engage AMPARs. The study further proposed that Noelin 1 promotes dimeric AMPAR assemblies, potentially contributing to receptor clustering and synaptic organization. Overall, this is no doubt an exciting, novel and important contribution to the iGluR field, the manuscript is well written, so I highly recommend its publication in Nature, subject to the following revisions and clarifications.

1. The authors report that over 30% of the native CP-AMPA receptor population purified from rat brain is in complex with Noelin 1. While this is intriguing and supports a potentially significant physiological role, it raises the question of whether this proportion accurately reflects the in vivo distribution or is influenced by selective stability during purification. The authors should discuss whether this could result from biochemical bias, such as if Noelin-bound receptors are more stable, less prone to dissociation, or preferentially retained during detergent extraction and cryo-EM sample preparation.
2. The manuscript would benefit from adding comparison between the native CP-AMPA receptor structures presented here and previously published structures of CI-AMPA receptors, particularly those containing edited GluA2. Highlighting similarities and differences in auxiliary protein occupancy, ATD conformation, and subunit stoichiometry would provide important context for understanding the function of CP-AMPA receptors.
3. While the authors provided structural and functional characterizations of Noelin 1 on AMPAR, the ephys characterization of Noelin-bound receptor is somewhat limited. To confirm that Noelin 1 doesn't affect AMPAR gating, more detailed kinetic analyses are required to definitively support this conclusion. I suggest the authors either tone down the conclusion or provide more detailed kinetic analysis.
4. The claim that Noelin 1 promotes receptor dimerization is primarily based on FSEC traces. While these results are interesting, they are not sufficient to establish direct dimerization or clustering. The authors should moderate the claim and discuss its current limitations which won't affect the novelty of the current work.
5. (Minor) The flowchart in Fig 1 is overly dense and the AMPAR receptor cartoon unnecessarily detailed. A simplified schematic highlighting the major purification steps and better contrast of color choices would improve clarity and better guide the readers through the experimental design.
6. The discussion of TARP and CNIH subunit occupancy is relatively brief and lacks comparison with available complex structures, particularly with existing CI-AMPA receptor structures to illustrate how auxiliary subunit stoichiometry and positioning

may differ across receptor subtypes.

7. In Fig. 3d, the various gray colors are confusing.

8. What do the shifts mean in Extended Data Fig. 6j?

9. Representative densities at the Noelin and GluA4 should be shown.

10. Line 28, the full term of this two acronyms CI and CP should be mentioned here.

Referee #2

(Remarks to the Author)

The study by Fang et al. presents cryo-EM data of native AMPARs isolated from rat cerebellum. This work builds upon the group's previous investigation of native AMPARs from hippocampus (Yu et al., Nature, 2021) and uses combinations of subunit-specific antibodies to purify the different receptor subtypes. The receptor classes in the cerebellum are known to be different from those of the hippocampus, exhibiting distinct proportions of the AMPAR subunits and auxiliary proteins.

The authors focussed on classes of GluA2-lacking calcium-permeable AMPARs (CP-AMPARs) which are expected to be prevalent in both specific glial cell populations and certain interneuron subtypes, and identify several different AMPAR combinations including GluA1/4 heteromers. As expected, these receptors are associated with auxiliary subunits, prominently the TARPs, but also CNIHs. Of great interest, Noelin 1 (Noe 1) was visualized bound to the GluA4 NTD, as such this represents the first structure of an AMPAR bound to an extracellular interactor.

Having identified the native combination with Noe 1, they moved on to investigate the GluA4-Noe 1 interaction at higher resolution by expressing the native combination GluA1/4 + Noe 1 in heterologous cells. This allowed them to identify the molecular details of the interaction and how it impacted on AMPAR structural behaviour, but without influencing receptor function. They ultimately speculate that due to the multimerization of Noelins, the interaction identified may represent a means by which receptors can be clustered at synapses.

This study will likely be of great interest in the fields of glutamate receptor and cerebellar research, and has wider implications for our understanding of synapses and synaptic plasticity.

I have just a few points:

1) The authors reasonably make the case that the receptors they have identified predominantly derive from Bergmann glia that express GluA1 and GluA4. Are they able to discern more about the nature of the auxiliary subunits found? Bergmann glia express $\gamma 4$ and $\gamma 5$ but may not necessarily have much $\gamma 2$. Are there any points of difference in the TM regions (Extended data Fig 4) that would point to the predominant TARPs?

2) While the authors are careful not to explicitly identify their CP-AMPARs as 'synaptic', the language used (particularly in relation to noelin) is strongly focussed on synaptic function. This creates something of a disconnect, as Bergmann glia do not receive conventional synaptic connections. In this context it would seem appropriate to cite the work of Matsui & Jahr (e.g. Matsui and Jahr 2003 or Matsui et al 2005) who suggest that there are "relatively uniform and modest densities of AMPA receptors throughout the BG membrane". that are activated by ectopic vesicular release.

3) I would find it useful supplementary information if an alignment of the GluA1-4 NTDs was included, with the Noelin interacting residues highlighted. It will be useful to build the idea of how the selectivity for GluA4 (and partial for GluA2) is achieved. At least some of the residues appear to be conserved, but it's difficult to track in Sup Fig 2 currently without an annotated alignment.

4) Line 96: oligodendrocyte precursor cells are not mentioned in Ref 34. Perhaps the authors meant to cite Zonouzi et al 2011.

Version 1:

Reviewer comments:

Referee #1

(Remarks to the Author)

The authors have fully addressed my comments/questions. I look forward to seeing this exciting work published in Nature. Congratulations to the team.

Referees' comments:

Referee #1 (Remarks to the Author):

1. *The authors report that over 30% of the native CP-AMPA population purified from rat brain is in complex with Noelin 1. While this is intriguing and supports a potentially significant physiological role, it raises the question of whether this proportion accurately reflects the in vivo distribution or is influenced by selective stability during purification. The authors should discuss whether this could result from biochemical bias, such as if Noelin-bound receptors are more stable, less prone to dissociation, or preferentially retained during detergent extraction and cryo-EM sample preparation.*

Reply. We appreciate this comment and understand that there are limitations to which we can extrapolate from the biochemical isolated CP-AMPA and the state of the CP-AMPA complexes in the intact rodent brain. We have therefore tempered our statements and have included the following sentence, on lines 242-244, "Nevertheless, we note that Noe 1 may stabilize the CP-AMPA complex, thus making it more abundant in the purified preparation in comparison to the relative abundance of the complex *in vivo*."

2. *The manuscript would benefit from adding comparison between the native CP-AMPA structures presented here and previously published structures of CI-AMPA, particularly those containing edited GluA2. Highlighting similarities and differences in auxiliary protein occupancy, ATD conformation, and subunit stoichiometry would provide important context for understanding the function of CP-AMPA.*

Reply. We appreciate the reviewer's insightful suggestion. In the revised manuscript, we have added a supplementary figure for published GluA2-containing CI-AMPA structures to Supplementary Fig. 4. We have included the following sentences on lines 351-356 in the revised version, "Therefore, in comparison to GluA2-containing AMPARs, CI- and CP-AMPA share a similar overall architecture, comprising three distinct layers and similar auxiliary subunit occupancy. However, they differ in subunit stoichiometry and ATD conformation (Fig. 2, Supplementary Fig. 4). In CI-AMPA, GluA2 consistently occupies the B/D positions and is associated with a stable ATD layer (Supplementary Fig. 4) while GluA4 occupies the B/D positions in CP-AMPA, either with a stable or unstable ATD configuration."

3. *While the authors provided structural and functional characterizations of Noelin 1 on AMPAR, the ephys characterization of Noelin-bound receptor is somewhat limited. To confirm that Noelin 1 doesn't affect AMPAR gating, more detailed kinetic analyses are required to definitively support this conclusion. I suggest the authors either tone down the conclusion or provide more detailed kinetic analysis.*

Reply. We have downplayed the conclusion that there are no effects on receptor kinetics and simply state that "we find that Noe 1 binding to CP-AMPA has no discernable effects on receptor **desensitization kinetics**. Thus, a synaptic scaffolding protein can restrain the conformational space of the ATD and LBD layers, **while having little** impact on receptor function", on lines 439-442.

We edited the text on line 357 to state that Noelin does not meaningfully alter receptor **desensitization kinetics**, rather than, more generally, AMPAR gating.

We would like to mention that Boudkkazi et al.'s analyses also did not reveal obvious changes in synaptic kinetic changes in Noe KO mice, despite large reductions in response amplitudes and that they too found that Noelin had no effect on the rates of GluA4 receptor deactivation or desensitization (PMID: 37591201), as cited in the main text on lines 372-373.

4. *The claim that Noelin 1 promotes receptor dimerization is primarily based on FSEC traces. While these results are interesting, they are not sufficient to establish direct dimerization or clustering. The authors should moderate the claim and discuss its current limitations which won't affect the novelty of the current work.*

Reply. We have moderated this claim by editing the text on lines 418-422 to read, "We **speculate** that Noe 1 **could** function as a dimerization agent and, upon interaction with additional synaptic proteins, may stabilize and organize AMPARs at the synaptic cleft⁴⁹, thereby enhancing AMPAR signaling. **However, we note that further studies, *in vivo*, are required to establish the role of Noe 1 in receptor dimerization or clustering.**"

5. (Minor) *The flowchart in Fig 1 is overly dense and the AMPAR receptor cartoon unnecessarily detailed. A simplified schematic highlighting the major purification steps and better contrast of color choices would improve clarity and better guide the readers through the experimental design.*

Reply. In line with this comment, we have replaced the flowchart in Fig. 1 with a simplified version, we have reduced the level of detail in the AMPAR cartoon to avoid distraction and to improve visual clarity. Furthermore, we have adjusted the contrast between elements to make the figure easier to interpret.

6. *The discussion of TARP and CNIH subunit occupancy is relatively brief and lacks comparison with available complex structures, particularly with existing CI-AMPA structures to illustrate how auxiliary subunit stoichiometry and positioning may differ across receptor subtypes.*

Reply. We thank the reviewer for the insightful suggestion. In the revised manuscript, we have expanded our discussion to include comparisons with previously published CI-AMPA structures on lines 182-188 of the main text, "Previous structural studies of GluA2-containing CI-AMPA structures have shown defined patterns of auxiliary subunit occupancy (Supplementary Fig. 4), that include four TARP γ 2 subunits in GluA2 homomers^{34,41}, two TARP γ 8 subunits in GluA1/A2 heteromers⁴², and various combinations of TARP γ 8 and CNIH2 subunits occupying distinct positions in native and recombinant receptors^{43,44}. In these complexes, TARPs are consistently found in the B'/D' positions, while there is greater auxiliary subunit heterogeneity at the A'/C' positions, similar to our native CP-AMPA structures."

7. *In Fig. 3d, the various gray colors are confusing.*

Reply. We have adjusted the color scheme to make the figure more readily interpretable.

8. *What do the shifts mean in Extended Data Fig. 6j?*

Reply. Extended Data Fig. 6j relates to Fig. 3d. For each GluA4 mutant, we measured the peak shift after incubation with dNoe1 based on these FSEC traces. This is the experimental data that is then summarized in the main text Fig. 3d. We have clarified this point by editing the legend of Extended Data Fig. 6j.

9. *Representative densities at the Noelin and GluA4 should be shown.*

Reply. We appreciate this comment and have added a panel to Extended Data Fig. 4 to show the densities at the Noelin and GluA4.

10. *Line 28, the full term of this two acronyms CI and CP should be mentioned here.*

Reply. Done.

Referee #2 (Remarks to the Author):

1) *The authors reasonably make the case that the receptors they have identified predominantly derive from Bergmann glia that express GluA1 and GluA4. Are they able to discern more about the nature of the auxiliary subunits found? Bergmann glia express $\gamma 4$ and $\gamma 5$ but may not necessarily have much $\gamma 2$. Are there any points of difference in the TM regions (Extended data Fig 4) that would point to the predominant TARPs?*

Reply. We appreciate this comment and have wrestled with the general notion raised by the reviewer. As suggested, we closely examined the TM regions of the auxiliary subunits in our cryo-EM maps for potential features indicative of specific TARP isoforms. However, due to the moderate local resolution of 4-5 Å of the auxiliary subunits and the high structural similarity among TARP family members, we were unable to reliably distinguish between individual isoforms. Complementary mass spectrometry analysis (Extended Data Fig. 1) revealed that $\gamma 2$ appears to be the most abundant auxiliary subunit, with $\gamma 4$, $\gamma 5$, $\gamma 7$, and $\gamma 8$ also detected in the sample. Future cryo-EM studies, at higher resolution, will provide more insight into the conclusive identification of specific TARP subunits.

2) *While the authors are careful not to explicitly identify their CP-AMPA receptors as 'synaptic', the language used (particularly in relation to noelin) is strongly focussed on synaptic function. This creates something of a disconnect, as Bergmann glia do not receive conventional synaptic connections. In this context it would seem appropriate to cite the work of Matsui & Jahr (e.g. Matsui and Jahr 2003 or Matsui et al 2005) who suggest that there are "relatively uniform and modest densities of AMPA receptors throughout the BG membrane". that are activated by ectopic vesicular release.*

Reply. We thank the reviewer for highlighting this point. We have now cited the work of Matsui and Jahr (2003) as suggested. We have further updated the main text as follows. Lines 90-91 "**CP-AMPA receptors, distributed throughout Bergmann glia, are activated by ectopic glutamate release onto those cells**^{31,32}." Lines 305-307, "Indeed, Bergmann glia AMPARs significantly contribute to fine-tuning **neuroglial association and neuronal processing**⁷ through sensing **glutamate released from ectopic synaptic sites**³²." Lines 443-446, "Through the interactions between tetrameric Noe 1 and the CP-AMPA receptors ATDs, we provide evidence of how Noe 1 can bind to two receptor complexes, thus providing insight into a possible mechanism underpinning Noe 1 clustering of synaptic receptors **and receptors at neuroglial positions.**"

3) *I would find it useful supplementary information if an alignment of the GluA1-4 NTDs was included, with the Noelin interacting residues highlighted. It will be useful to build the idea of how the selectivity for GluA4 (and partial for GluA2) is achieved. At least some of the residues*

appear to be conserved, but it's difficult to track in Sup Fig 2 currently without an annotated alignment.

Reply. We appreciate the reviewer's comment. As shown in Extended Data Fig. 7a, we provide a sequence alignment of rat GluA1-A4 highlighting the residues within the Noelin-GluA4 interaction interface. In the revision, we added a Supplementary Fig. 7 showing structural alignments of the GluA1-4 at Noe 1 interfaces.

4) Line 96: *oligodendrocyte precursor cells are not mentioned in Ref 34. Perhaps the authors meant to cite Zonouzi et al 2011.*

Reply. Thank you. The reference has been corrected.